# *Xanthomonas* effector XopR hijacks host actin cytoskeleton via complex coacervation

He Sun[1], Xinlu Zhu[1], Chuanxi Li[2], Zhiming Ma[1], Xiao Han[1], Yuanyuan Luo[1], Liang Yang [1,3], Jing Yu [2] & Yansong Miao [1✉]

The intrinsically disordered region (IDR) is a preserved signature of phytobacterial type III effectors (T3Es). The T3E IDR is thought to mediate unfolding during translocation into the host cell and to avoid host defense by sequence diversification. Here, we demonstrate a mechanism of host subversion via the T3E IDR. We report that the *Xanthomonas campestris* T3E XopR undergoes liquid-liquid phase separation (LLPS) via multivalent IDR-mediated interactions that hijack the Arabidopsis actin cytoskeleton. XopR is gradually translocated into host cells during infection and forms a macromolecular complex with actin-binding proteins at the cell cortex. By tuning the physical-chemical properties of XopR-complex coacervates, XopR progressively manipulates multiple steps of actin assembly, including formin-mediated nucleation, crosslinking of F-actin, and actin depolymerization, which occurs through competition for actin-depolymerizing factor and depends on constituent stoichiometry. Our findings unravel a sophisticated strategy in which bacterial T3E subverts the host actin cytoskeleton via protein complex coacervation.

[1] School of Biological Sciences, Nanyang Technological University, Singapore, Singapore. [2] School of Materials Science and Engineering, Nanyang Technological University, Singapore, Singapore. [3] Singapore Centre for Environmental Life Sciences Engineering, Nanyang Technological University, Singapore, Singapore. ✉email: yansongm@ntu.edu.sg

The type III secretion system (T3SS) is the primary virulence factor of phytobacterial pathogens for plant invasion through injecting type-III effectors (T3Es). The mechanisms by which T3Es hijack host cellular processes are diverse and profoundly complex. During host–pathogen coevolution, bacterial T3Es develop a large number of intrinsically disordered regions (IDRs) that increase genetic variation, enable diversified targeting of host biomolecules, and evade sequestration by host-evolved guard or decoy systems[1–4]. While IDRs are known to provide the structural flexibility that facilitates the passing of T3Es through the narrow channel of the T3SS[5], the underlying mechanisms by which bacterial IDRs subvert host biology remain elusive. Due to the abundant disordered regions, different T3Es show low conservation in sequence. However, they often strategically target and subvert a narrow spectrum of cellular processes.

The plasma membrane (PM)-cortical actin cytoskeleton (AC) continuum is one of the host systems most commonly hijacked by T3Es, occurring after injection and before the neutralization of T3E by host surveillance molecules[6]. More than 30% of bacterial T3Es target the cell membrane of eukaryotes, while many others subvert or exploit the AC[7,8]. Actin polymerization and depolymerization are precisely orchestrated by multiple actin-binding proteins (ABPs), including actin nucleators, cross-linkers, and actin-depolymerizing factors (ADFs)[9]. Phytopathogenic bacteria were shown to induce time-dependent remodeling of plant cortical actin arrays at different stages of infection. While early infection triggered an increase in plant F-actin production via pathogen-associated molecular pattern (PAMP)-triggered immunity (PTI), T3SS played essential roles in actin bundling in the plant at the late stage of infection[10,11]. However, the molecular mechanism by which phytobacterial effectors remodel the plant AC has remained enigmatic. Whether the evolutionarily preserved IDRs of T3E play roles in host actin remodeling is not known. Growing evidence shows that IDRs mediate a network of weak- and multivalent-interactions that could be coupled with liquid-liquid phase separation (LLPS) to regulate diverse cellular processes, such as actin assembly, during signal transduction[1,12]. Membrane-integrated or membrane-associated signaling proteins and ABPs were able to generate nano- or mesoscale liquid-like clusters on the 2D fluid surface surrounded by 3D cytoplasmic space. During T-cell signaling, the activities of the PM-associated actin nucleation factor Nck/WASP and Ras are modulated by varying ligand engagement by tuning the dwelling time and stoichiometry of the biomolecules in the biochemical compartment[13–15].

In this work, we identify a T3E, XopR, from the phytobacteria *Xanthomonas campestris* pv. *Campestris (Xcc)*. XopR has abundant disordered residues and contains potential PM- and actin-binding motifs. Our quantitative cell biology, biochemistry, and biophysical experiments have revealed that XopR forms 2D-complex coacervation for actin remodeling via IDR on the Arabidopsis PM. During T3SS-mediated translocation, XopR undergoes progressive complex coacervation by interacting with PM-localized actin nucleator type I formin and cortical F-actin underneath the Arabidopsis PM. The chemical–physical properties of XopR coacervates are finetuned for electrostatic- and stoichiometry-dependent remodeling of the plant AC. XopR manipulates actin nucleation by controlling formin clustering, crosslinks cortical F-actin via multivalent scaffolding, and antagonizes the function of ADF by competing for F-actin binding. Overall, we show that IDR-containing T3E remodels the plant AC through multifaceted, multivalent interactions with host ABPs during bacterial invasion. This work deciphers an attack mechanism during pathogen–host interactions, in which T3Es hijack and subvert the host AC by forming complex biomolecular coacervates.

## Results

### PM-targeted type III effector (T3E) XopR remodel *Arabidopsis* AC during *Xcc* infection.

We first performed a sequence analysis of *Xanthomonas campestris* pv. *Campestris (Xcc)* effectors to identify the potential surface-anchoring effectors that may target the host plasma membrane-actin cytoskeleton (PM-AC) continuum. Among the four effectors with predicted membrane-anchoring amphipathic alpha-helices, we focused on XopR, which had the highest percentage of intrinsically disordered residues (Supplementary Fig. 1a). We first tested XopR secretion by *Xcc* during infection and its localization in the host by utilizing a split GFP system[16] to monitor its live-cell translocation in Arabidopsis after *Xcc* inoculation. Here, we tagged the C-terminus of XopR in the *Xcc* genome with GFP11 (the 11th β strand with 13 amino acids). Engineered XopR-GFP11-containing *Xcc* was used to flood-inoculate 7-day-old *Arabidopsis* plants stably expressing GFP1-10 (1–10 β strands) (Supplementary Fig. 1b). During the first 24 h of *Xcc* infection, XopR-GFP signals gradually increased and started to show weak PM localization at 6 h postinoculation (hpi) and a clear PM pattern at 12 hpi (Fig. 1a, b), indicating the cumulative delivery of XopR into host cells. The PM localization of XopR during real-time *Xcc* injection in the physiological context was reminiscent of the PM localization of exogenously overexpressed *Xanthomonas oryzae* pv. *oryzae* XopR in the plant protoplast[17]. Next, we investigated whether PM-targeted XopR changes cortical F-actin in the host. We quantitatively characterized the changes in F-actin production and bundling in Arabidopsis using Lifeact-Venus-expressing seedlings by *Xcc* infection in the presence or absence of XopR[18,19]. The quantity of total F-actin production and bundling were quantified at 6-, 12-, and 24 hpi, by measuring the total signal intensity and skewness[10,11,20] of Lifeact-Venus-labeled filaments. Starting at 6 hpi, a noticeable increase in overall actin polymerization was observed without significant changes in bundling (Fig. 1c, d and Supplementary Fig. 1c). The total F-actin density continued to increase from 6 to 12 hpi, which was significantly contributed by the XopR function (Fig. 1c, d and Supplementary Fig. 1c). Interestingly, starting at 12 to 24 hpi, a significant burst of F-actin bundling was observed, in which T3SS was indispensable and XopR played a substantial role. *Xcc*-stimulated actin-bundling was drastically attenuated when using *XccΔxopR* (Fig. 1c, e and Supplementary Fig. 1d) and disappeared when using the T3SS mutant *XccΔhrcC* (Supplementary Fig. 1e, f). The above results suggest that XopR manipulates host F-actin turnover functions differently at different stages of infection, including enhanced polymerization or reduced depolymerization during the early phase (6–12 hpi) and increased F-actin crosslinking during a later phase with effector secretion (12–24 hpi).

### *Xcc* XopR modulates *Arabidopsis* actin nucleation in vivo.

To better examine the changes in actin nucleation induced by XopR injection in the plant, we designed a Latrunculin B (LatB) washout assay and monitored the regeneration of actin seeds in *Xcc*-infected Arabidopsis. WT *Xcc* and *XccΔxopR* were first applied to Lifeact-venus-expressing seedlings for different durations. Actin filaments were then depolymerized entirely by 5 μM LatB for 30 min. Subsequently, immediately after LatB removal, Lifeact-Venus was imaged over time to monitor the reinitiation of actin polymerization. In uninfected *Arabidopsis*, short F-actin seeds were regenerated, starting at 30 min and continuing to increase over 1 h. Interestingly, Xcc-infected *Arabidopsis* exhibited differences in efficacy in reinitiating F-actin. At 12 hpi, *Arabidopsis* showed elevated actin repolymerization compared to the uninfected seedlings, whereas at 24 hpi, Arabidopsis demonstrated a noticeable delay in regenerating F-actin (Fig. 1f, g). By contrast, with

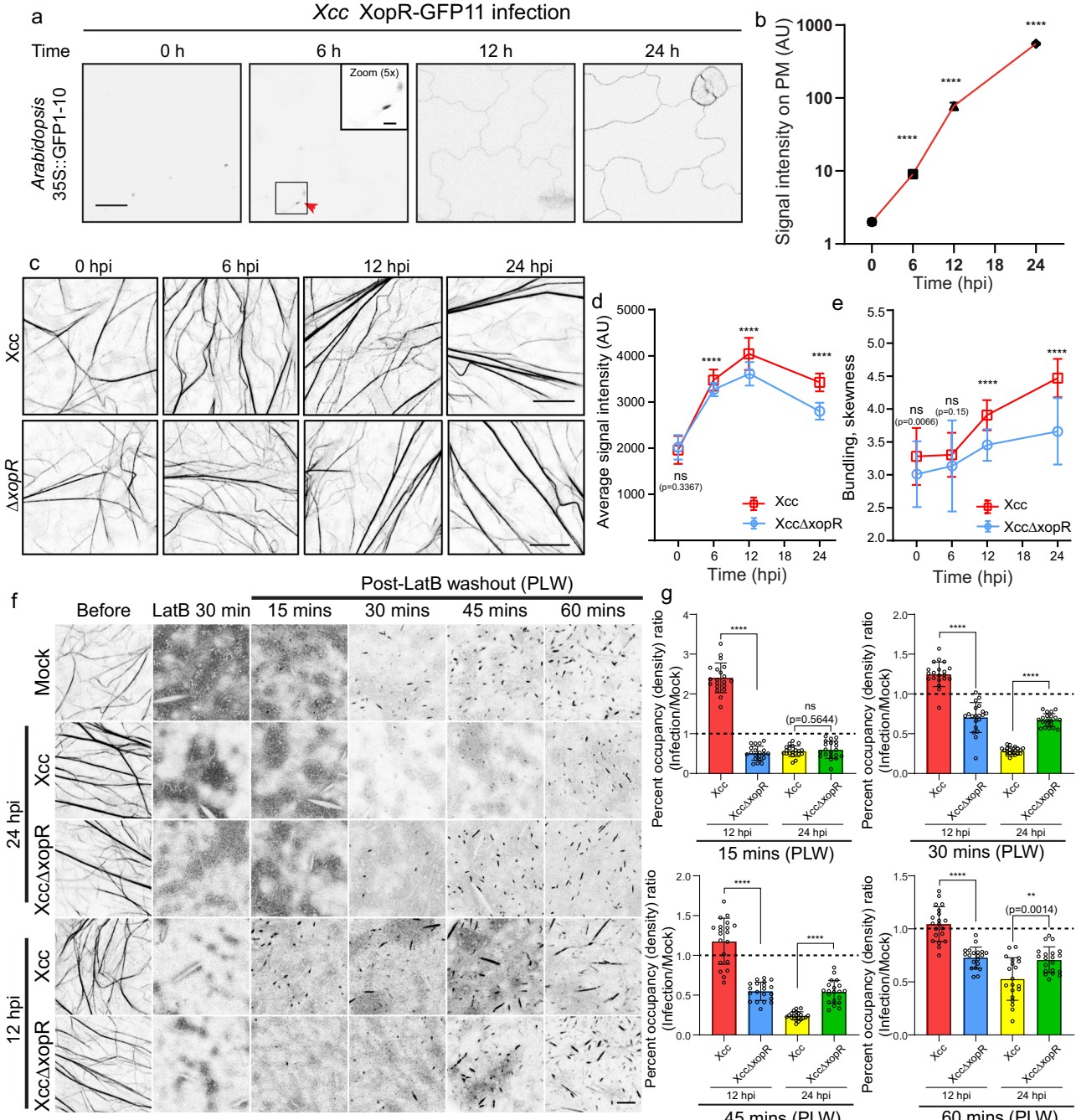

**Fig. 1 T3E XopR remodels the *Arabidopsis* actin cytoskeleton during *Xcc* infection. a, b** Representative images of XopR-GFP (self-complementing) at different time points and quantification ($n = 20$) after infection. The red arrow indicates weak signal accumulation on the plasma membrane. Data were presented as mean values ± SD. **c–e** Representative images of Lifeact-Venus in epidermal cells of *Arabidopsis* cotyledons and image quantification. Seven-day-old seedlings were dip-inoculated with WT *Xcc* and *Xcc*Δ*xopR*. Images were taken at 0, 6, 12, and 24 h postinoculation (hpi). The average signal intensity per image and skewness of Lifeact-Venus were measured ($n = 50$ images from ten individual seedlings). Data were presented as mean values ± SD. **f, g** Representative images and actin density analysis of Lifeact-venus in LatB washout assay. Seven-day-old seedlings were flood-inoculated with *Xcc* or *Xcc*Δ*XopR* at the indicated hpi, then subjected to 5 μM LatB treatment for 30 min before washout and image acquisition at the indicated time points of post-LatB washout (PLW). Percent occupancy was measured in the LatB washout assay (**g**, $n = 20$, from five seedlings). Data were presented as mean values ± SD. Two-tailed Student's *t*-test was performed assuming equal variance. Ns no significant difference, *$p < 0.05$, **$p < 0.01$, ***$p < 0.001$, ****$p < 0.0001$. AU arbitrary unit. Scale bar: 20 μm in **a** 5 μm in zoomed-in image of **a**, 10 μm in **c**, and 5 μm in **f**.

*Xcc*Δ*xopR* infection, evident desensitization for actin reinitiation was observed. Compared to WT *Xcc*, F-actin seeds were less-stimulated at 12 hpi and less-inhibited at 24 hpi in *Xcc*Δ*xopR*-infected seedlings (Fig. 1f, g), suggesting that XopR hijacks actin nucleation. Other *Xcc* virulence factors might also participate,

additively or synergistically, because *Xcc*Δ*xopR* did not entirely lose its ability in remodeling *Arabidopsis* actin (Fig. 1f, g). To investigate XopR-mediated host actin manipulation without introducing other bacterial virulence factors, such as PAMPs, we next generated stable transgenic Arabidopsis that expresses 35 S::Lifeact-

venus and XVE::XopR-mRuby2-FLAG. The XVE promoter enables the time-dependent induction of XopR upon the addition of β-estradiol (Supplementary Fig. 1g) to mimic the gradual secretion and accumulation of T3E in the host. Early-stage β-estradiol induction (~6 h) produced relatively weak XopR expression; however, the expression began to significantly enhance actin polymerization, as shown in the LatB washout assay (Supplementary Fig. 1g–i). Such early induction enhanced actin polymerization was abolished if the formin inhibitor SMIFH2 was applied for 3 h within the first 6 h of XVE beta-estradiol induction (Supplementary Fig. 1h, i), suggesting a formin-dependent enhancement of actin assembly by expressing XopR at a low level. However, XopR-induced actin reestablishment was largely attenuated once XopR was expressed at a higher level after 12 h or 24 h of induction (Supplementary Fig. 1g–i), indicating biphasic regulation of actin polymerization in an XopR dose-dependent manner. Such biphasic regulation is highly consistent with XopR-mediated actin remodeling under *Xcc* infection (Fig. 1f, g). Overall, XopR accumulation in Arabidopsis by either bacterial injection or inducible expression exhibited time- and dose-dependent subversion of the host AC.

**XopR undergoes LLPS via electrostatic interactions**. We next analyzed protein sequences to identify potential physical–chemical properties that are relevant to actin polymerization. We compared all the XopR homologs among *Xanthomonas* species to search for lipid- and actin-interactive motifs and protein–protein interaction domains. We found that all XopR orthologs are positively-charged overall and possess an N-terminal IDR, an amphiphilic helix, and a Wiskott–Aldrich homology 2 (WH2)-like motif (Fig. 2a and Supplementary Fig. 2a–c). The IDR often guides multivalently interactive molecular assembly though weak inter- and intraprotein interactions and may undergo LLPS when both dense-phase and dilute-phase coexist[12,21,22]. We first examined the self-interaction of XopR via surface plasmon resonance (SPR) using recombinant XopR protein (Supplementary Fig. 3a). The SPR sensorgram showed a continuous increase in self-association with increasing concentrations of XopR, as well as biphasic dissociation kinetics (Fig. 2b), suggesting the existence of high stoichiometric equilibria for XopR self-assemblies[23]. XopR is positively charged (pI = 10.77) with a net positive charge of +23.8 at neutral pH (Fig. 2b and Supplementary Fig. 2a), which motivated us to test whether the surface charge of XopR regulates its valency by comparing the elution profile of XopR from size exclusion chromatography at different ionic strengths. XopR was eluted as monomers at 300 mM NaCl, trimers at 150 mM NaCl saline, and multivalent oligomers at 50 mM NaCl (Fig. 2c). Interestingly, the XopR solution showed turbidity in 50 mM NaCl solution (Fig. 2e), an indication of either precipitation or the formation of lipid droplets in LLPS via multivalent interaction. Both light and electron microscopic examinations revealed that XopR proteins form spherical droplets (Fig. 2d and Supplementary Fig. 3b) that undergo coalescence (Supplementary Movie 1), a typical LLPS phenomenon. The phase behavior of XopR was further characterized by fluorescence imaging, in which the LLPS depended on both the XopR concentration and electrostatic strength of the solution (Fig. 2e). XopR coacervation is reversible, where XopR droplets are slowly dissolved either by reducing the protein concentration via dilution or increasing the ionic strength of the solution (Supplementary Fig. 3c). The reversibility of XopR droplets suggests a re-equilibration of XopR between dense and dilute phases by crossing over the phase boundary. Furthermore, we found that XopR LLPS is primarily dependent on the N-terminal IDR (N-IDR) but not the C-terminal folded region. XopR N-IDR

recapitulates a phase behavior similar to that of full-length XopR (Supplementary Fig. 3d–f).

Quantitative determinations of the adhesiveness and surface tension of coacervates are critical to an in-depth understanding of how the material properties tune the functions of biocondensates. To study the two-dimensional-associated XopR coacervates, we utilized a surface forces apparatus (SFA)[24] to quantitatively determine the adhesion force ($F_{ad}$) (Supplementary Fig. 3g). $F_{ad}$ results from the interfacial tension as a function of distance ($D$) between negatively charged mica (with radius $R$) surfaces under different ionic strengths. The $F_{ad}$ of XopR coacervates decreased with increasing salt concentration (Fig. 2f). An average $F_{ad}$ of −5.11 mN/m was detected in 50 mM NaCl saline, which was noticeably reduced to −1.22 mN/m in 100 mM NaCl. The decrease in $F_{ad}$ is likely due to the increase in the screening effect by the electrostatic interaction between XopR molecules. The ionic strength–responsive interaction force suggested that inter- and intra-electrostatic interactions are the driving force for the LLPS of XopR. We also tested whether counterion interactions between XopR and other negatively charged binding partners regulate the physical properties of the electrostatically tunable XopR coacervates. We introduced XopR with a supercharged mutant GFP that has a net surface charge of −30 [scGFP(−30)][25] to mimic negatively charged biomolecules. Interestingly, ScGFP(−30) and XopR undergo complex coacervation in an ionic strength-dependent manner (Supplementary Fig. 3h, i). XopR-scGFP(−30) coacervates exhibited greater adhesion forces ($F_{ad}$ = −12.9 mN/m) and surface tension than XopR alone (Supplementary Fig. 3j), although they still have a low interfacial energy of ~1 mN/m at 50 mM NaCl, similar to many complex coacervation systems, such as the polylysine (PLys)-polyglutamic acid (PGA) complex[26] and mussel adhesive protein (fp-151-RGD) with hyaluronic acid[27]. The above result suggested that scGFP(−30) enhanced the valency within the complex coacervates, likely by forming more interactions between oppositely charged patches of scGFP(−30) and XopR. Nevertheless, we could not exclude other potential interactions, such as those derived from the positively charged patches of XopR and possible pi–pi interactions.

**XopR coacervates with Arabidopsis type I formin in vitro**. Given the PM-association of XopR and its roles in initiating F-actin, we investigated whether XopR modulates the PM-localized formin actin nucleator and whether and how the molecular condensation of XopR would tune the behaviors and activities of host type I formins, which are integral membrane proteins with a single transmembrane domain[28]. We first tested this hypothesis biochemically by characterizing the interaction of XopR and a well-studied Arabidopsis AtFH1, as a representative of plant type I formin[29]. The AtFH1-FH1C domain (430–999 aa)[30] was produced from the prokaryotic expression system (Supplementary Fig. 3k). Strikingly, we found that AtFH1-FH1C displayed direct binding with XopR in vitro using the SPR system, in which the SPR sensorgram revealed a bivalent interaction mode (Fig. 2g)[23]. In the presence of XopR, AtFH1-FH1C demixed from the aqueous phase and concentrated in the same droplets as XopR in 50 mM NaCl solution, but this phenomenon occurred to a much lesser degree in 150 mM NaCl, indicating complex coacervation of XopR and AtFH1 (Fig. 2h and Supplementary Fig. 3l). In contrast to the drastic change in the effective interfacial energy (EIE) of XopR droplets upon the addition of scGFP(−30), XopR-AtFH1-FH1C coacervates displayed an EIE similar to that of XopR coacervates over a range of examined ionic strengths from 50 to

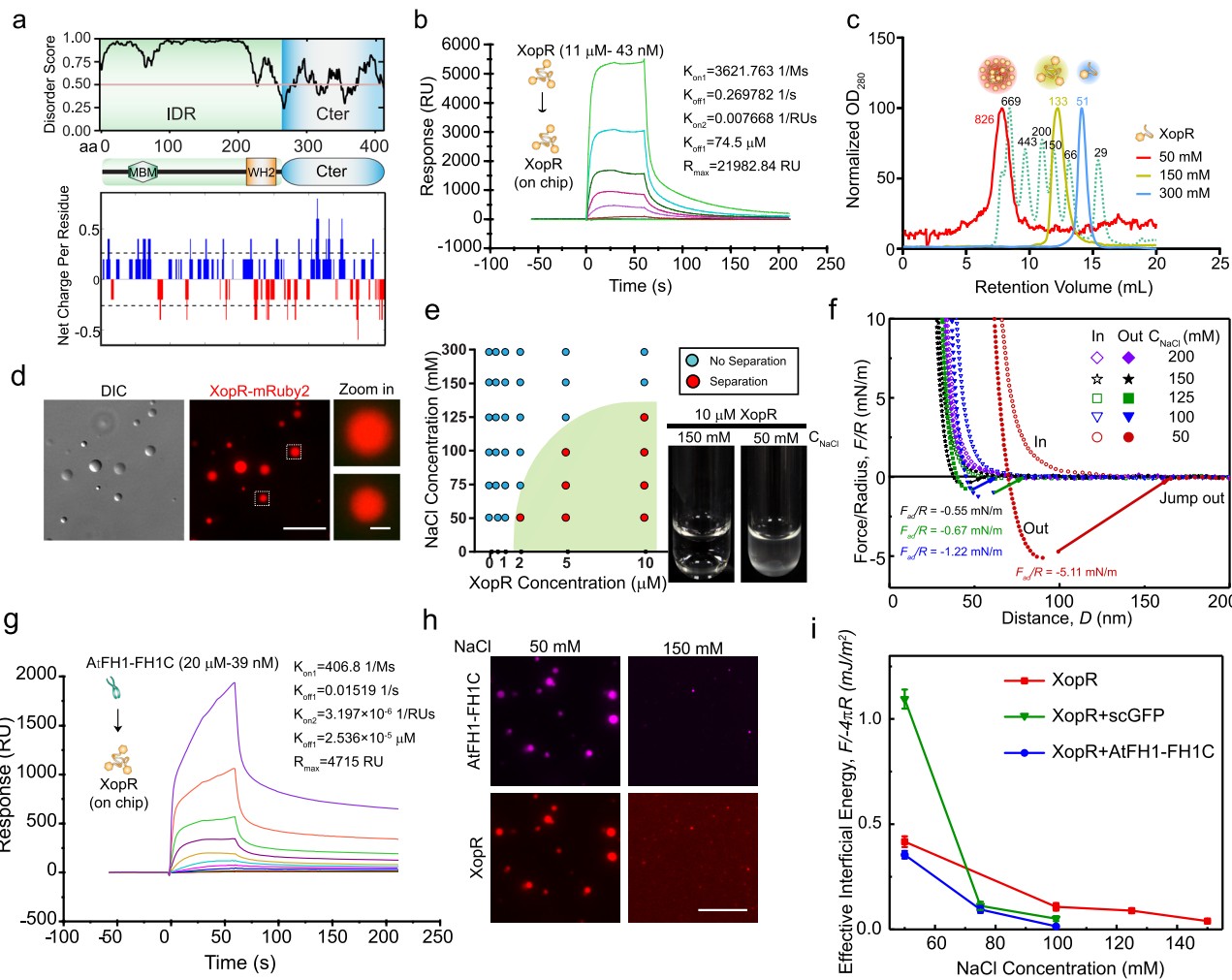

**Fig. 2 Biochemical and biophysical characterizations of XopR LLPS. a** Schematic diagram of the domains and charge pattern of XopR. IDR (upper panel) and charged residues (bottom panel) were analyzed by IUPRED2 and CIDER, respectively. **b** Representative SPR sensorgram of the XopR–XopR interaction. XopR at concentrations of 11 µM to 43 nM flowed over the chip with immobilized XopR. Binding parameters were generated using bivalent model. **c** Size exclusion chromatography of XopR at the indicated concentration of NaCl solution using Superdex 200 GL 10/300 Increase. The green dashed curve represents the elution profile of standard protein markers. **d** Liquid-liquid phase separation (LLPS) of XopR. XopR (10 µM, 10% XopR-mRuby2) was prepared in 50 mM NaCl solution, pH = 7.4, for 10 min before imaging. **e** XopR phase diagram was generated using ten images for each condition. Turbidity tests of XopR in solution are shown at the indicated NaCl concentration. **f** Typical force-distance profiles measured during approaching (open symbols) and separation (solid symbols) of the mica surfaces with an injected mixture of XopR coacervate as a function of the concentration of NaCl. **g** Representative SPR sensorgrams for XopR (on-chip) and AtFH1-FH1C, which were injected at concentrations of 20 µM to 39 nM. Binding parameters were generated using bivalent model. **h** Complex coacervation of XopR-AtFH1-FH1C in the low salt buffer (20 mM HEPES, 50 mM NaCl, pH = 7.4) and physiological buffer (20 mM HEPES, 150 mM NaCl, pH = 7.4). XopR (5 µM, 10% XopR-mRuby2) and AtFH1-FH1C (5 µM, 10% Alexa647-AtFH1-FH1C) were mixed for 10 min before imaging. **i** Effective interfacial energy of coacervates of XopR (10 µM), XopR-ScGFP (10 µM XopR + 10 µM ScGFP), and XopR-AtFH1-FH1C (10 µM XopR + 5 µM AtFH1-FH1C), as a function of NaCl concentration. Each of the effective surface energy values is averaged from three measurements. Scale bar: 10 µm in **d**, 1 µM for magnified images in **d**, and 10 µm in **h**.

100 mM NaCl (Fig. 2i and Supplementary Fig. 3m). With increasing NaCl concentration, XopR-scGFP(−30) demonstrated a faster decline in EIE than both the XopR and XopR-AtFH1 coacervates (Fig. 2i and Supplementary Fig. 3i, j). This result suggested that the intramolecular interaction of XopR and the intermolecular interaction between XopR and scGFP (−30) synergized with the intricate scaffolding of XopR-scGFP (−30) coacervates, in which electrostatic forces are critical for both types of interactions. However, the macromolecular assemblies of XopR-AtFH1 and AtFH1-FH1C did not create additional contacts for the multivalent network[21] and were less likely to be electrostatic-based interactions. Therefore, they did not change the sensitivities to electrostatic perturbation and did not increase the surface tension of XopR coacervates.

**XopR clusters PM formins and tunes formin activities in actin nucleation via molecular condensation.** The direct interaction between XopR and AtFH1-FH1C motivated us to examine their dynamic interactions in vivo. Given that the 35 S::AtFH1-GFP stable transgenic line is lethal, we examined type I formin regulation by XopR using the Arabidopsis type I formin line, 35 S::AtFH6-GFP[28,31,32], which shows structural conservation with AtFH1 and generates detectable signals under variable angle-total internal reflection fluorescence microscopy (VA-TIRFM). We first validated the physical interaction between XopR and AtFH6-FH1C biochemically via SPR by injecting XopR onto the SPR chip coated with the recombinant AtFH6-FH1C domain (294–899 aa) (Supplementary Fig. 4c, d). Then, we monitored AtFH6-GFP behavior at the single-particle level in *Arabidopsis* seedlings using

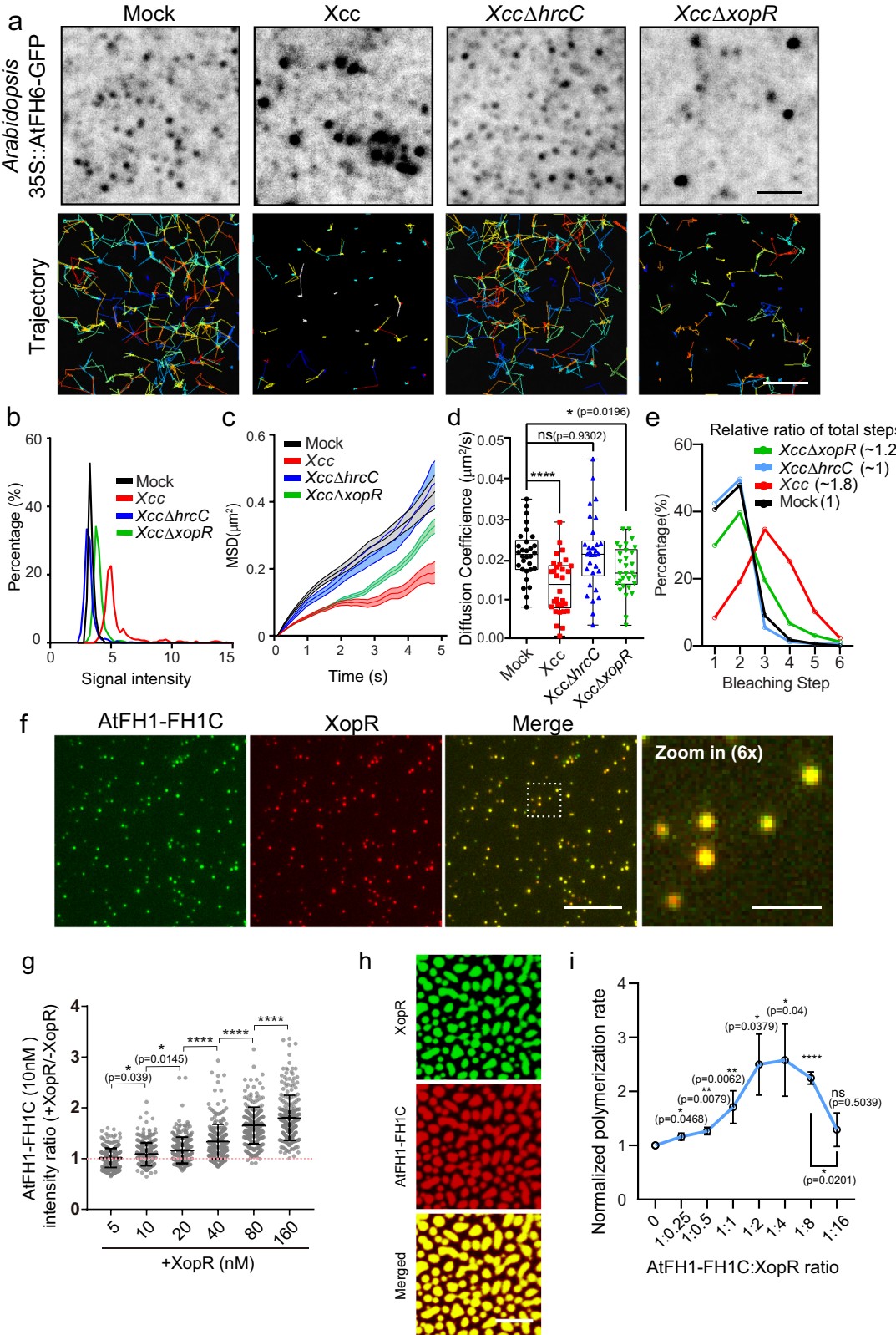

VA-TIRFM. At 24 h after *Xcc* inoculation, a large portion of AtFH6-GFP proteins was concentrated into brighter and larger nanoclusters from the small puncta that were displayed in the uninfected Arabidopsis (Fig. 3a, b). Via single-particle tracking and rheology analysis, AtFH6-GFP puncta showed heterogeneous dynamics on the cell surface. The primarily diffusive movement of AtFH6-GFP in untreated seedlings became a confined

diffusion mode at 24 hpi (Fig. 3a–d). However, the *Xcc*-infection-triggered clustering and immobilization of AtFH6-GFP were both attenuated when using *XccΔxopR* and abolished with *XccΔhrcC* (Fig. 3a–d). To evaluate the multivalent interactions of *Xcc*-triggered formin nanoclusters, we applied a single-particle photobleaching assay that enables the measurement of low-order protein oligomerization[33,34]. The mean step size of

**Fig. 3 XopR clusters Arabidopsis formin in vivo and in vitro. a** Representative images of the AtFH6-GFP clusters and moving trajectories in *Arabidopsis*. Seven-day-old seedlings were dip-inoculated with *Xcc/XccΔhrcC/XccΔxopR* for 24 h before imaging using VA-TIRFM. **b–e** Distributions of signal intensity (from left to right, $n = 454$, $n = 406$, $n = 650$, and $n = 636$ punctates), mean square displacement (MSD) and diffusion coefficient ($n = 0$ movies from six seedlings for each infection assay), and percentage distribution of the bleaching step ($n = 165$ punctates for mock, $n = 167$ for *Xcc*, $n = 165$ for *XccΔhrcC*, and $n = 164$ for *XccΔxopR*) were analyzed for AtFH6-GFP foci. The relative ratios of total bleaching steps are indicated in brackets, which were normalized by mock without *Xcc*. Error bands and error bars in Fig. 3c, d are ± SD. Whiskers represent min to max. **f** Representative dual-color TIRF images of 2.5 nM AtFH1-FH1C (10% Alexa647-AtFH1-FH1C) with 2.5 nM XopR (10% XopR-mRuby2) on an immobilized supported lipid bilayer (SLB). **g** Signal intensity quantification of AtFH1-FH1C (50% Alexa647 labeled) on SLB for Supplementary Fig. 5g ($n = 250$ particles, Error bar, SD). **h** Representative dual-color confocal images of AtFH1-FH1C (5 μM, 10% Alexa647-AtFH1-FH1C) with XopR (5 μM, 10% Alexa488-XopR) that were incubated on dynamic SLB with 50 mM NaCl for 15 min before imaging. **i** Actin polymerization rate in the pyrene–actin assay, which was normalized by spontaneous actin polymerization, in the presence of 100 nM AtFH1-FH1C and XopR (left to right, 0, 25, 50, 100, 200, 400, and 1600 nM) at the indicated stoichiometries in the presence of 5 μM profilin AtPRF1 ($n = 4$ for 0, 25, 50, 100, 200 nM, $n = 6$ for 400 nM, and $n = 7$ for 1600 nM; Error bar, SD). Scale bar: 2 μm for **a**, 10 μm for **b–f**, 2 μm for magnified images in **f**, 10 μm for **h**. Two-tailed Student's *t*-test was performed assuming equal variance. Ns no significant difference, *$p < 0.05$, **$p < 0.01$, ***$p < 0.001$, ****$p < 0.0001$.

photobleaching traces was measured, and the majority of AtFH6-GFP puncta showed one- or two-step sizes in untreated Arabidopsis (Fig. 3e), suggesting intrinsic heterogeneity with a low number of subunits. Strikingly, the subunit number of AtFH6 was drastically enhanced in the cluster at 24 hpi after *Xcc* inoculation (Fig. 3e and Supplementary Fig. 4a, b). In contrast, *XccΔhrcC* and *XccΔxopR* showed a much less induction of higher-order clustering of AtFH6-GFP (Supplementary Fig. 4a, b). Given the dimer as the minimum functional unit of formin[35–38], the above results suggest that XopR recruits and condenses formin dimers into surface nanoclusters. In addition, we observed that the subunits of formin in nanoclusters increased slightly after 6 hpi, which seemed to be independent of T3SS and XopR (Supplementary Fig. 4a, b), implying that the other *Xcc* virulence factors, such as PAMP, might stimulate formin nanoclustering during early infection. We next generated transgenic Arabidopsis expressing XVE::XopR-mRuby2 and AtFH6-GFP and examined in vivo formin clustering by expressing XopR gradually without introducing other bacterial virulence factors. Interestingly, during the induction of XopR by β-estradiol, surface AtFH6-GFP proteins were condensed into bright clusters at 12 h and highly colocalized with XopR-mRuby2, which grew over time and generated larger, immobilized amorphous patches at 24 h (Supplementary Fig. 4e–i). However, an IDR-deleting XopR variant, XVE::XopR-C-mRuby2, could not induce AtFH6-GFP clustering or the stabilization of AtFH6-GFP on the cell surface (Supplementary Fig. 4j–m).

To test whether the nanoclustering of formins and their in vivo colocalization with XopR depended on their physical interactions, we next reconstituted XopR-mediated AtFH6 clustering on an artificially supported lipid bilayer (SLB) (Supplementary Fig. 5a). Recombinant AtFH6-FH1C was first incorporated onto the SLB via DGS-NTA(Ni⁺) and their lateral motility and oligomerization were quantitatively examined. Upon supplementation with XopR, acute immobilization of AtFH6-FH1C-GFP on the SLB was observed (Supplementary Fig. 5b–d), which was consistent with the in vivo AtFH6-GFP immobilization after overexpressing XopR (Supplementary Fig. 4e–i). Next, we sought to characterize the initial formin clustering upon XopR injection during earlier infection using an SLB-based reconstitution system. We performed single-particle photobleaching of stabilized formin molecules on the SLB for step size counting over time. We purposely inhibited the lateral dynamics of the lipid molecules of the SLB by increasing the final concentration of DGS-NTA(Ni⁺)[39]. Membrane fluidity was examined via fluorescence recovery after photobleaching (FRAP) experiments. While DGS-NTA(Ni⁺) at less than 5% did not noticeably change membrane fluidity, a higher concentration starting from 10% inhibited SLB fluidity (Supplementary Fig. 5e). AtFH1-FH1C and XopR colocalized

highly on SLB into the same punctate spots (Fig. 3f). To mimic the cumulative secretion of XopR, we applied increasing doses of XopR to a fixed concentration of AtFH1-FH1C on the SLB. Dynamic fusion of AtFH1-FH1C puncta on the SLB was observed upon the addition of XopR (Supplementary Fig. 5f). The incubation of AtFH1-FH1C with incremental increases in XopR revealed gradual enhancement of formin signal intensity (Fig. 3g and Supplementary Fig. 5g) and photobleaching steps of AtFH1-FH1C-GFP in a stoichiometry-dependent manner (Supplementary Fig. 5h–j). Starting from 50 nM XopR at a XopR-formin stoichiometry of 1:5, formins showed a clear upshift in clustering, which indicates the approximate threshold concentration (Supplementary Fig. 5j). However, increasing the concentration of XopR will eventually generate large micron-sized clusters (Supplementary Fig. 5g). To examine the LLPS behaviors of AtFH1 and XopR on the lipid bilayer, we used higher concentrations of 5 μM AtFH1-FH1C-GFP and XopR, both of which join the same dense phase on the two-dimensional SLB (Fig. 3h and Supplementary Movie 2). The sizes of the SLB-based AtFH1-XopR assemblies are dependent on ionic strength (Supplementary Fig. 5k), which is consistent with electrostatic-mediated complex coacervation in solution (Fig. 2h). The 2D-LLPS of XopR-formin on SLB is reminiscent of the formin-XopR condensates in vivo (Supplementary Fig. 4e), which are, however, amorphous in shape, indicating the involvement of other traction forces at the cell surface that create complex connectivity for XopR-formin condensates in vivo.

We next investigated whether and how formin activity in actin polymerization is modulated by XopR coacervation. We applied different doses of XopR to a fixed concentration of AtFH1-FH1C in a range of molar ratios from 0.25:1 to 16:1. AtFH1-mediated actin polymerization was first investigated via pyrene–actin assay under physiological concentrations with 150 mM NaCl and in the presence of profilin AtPRF1. Interestingly, we found that AtFH1 activities in actin nucleation were regulated differently by XopR at two phases in a stoichiometry-dependent manner. In the stoichiometry range of 0.25:1 to 8:1 of XopR-AtFH1-FH1C, the initial polymerization rate of F-actin by formin was elevated significantly, but decreased when the stoichiometry was increased to 16:1 (Fig. 3i and Supplementary Fig. 6a–c). Notably, XopR and AtRPF1 did not exhibit obvious interactions when tested via SPR (Supplementary Fig. 6d). To further dissect XopR-mediated formin nucleation, we applied the TIRF-actin polymerization assay to measure the nucleation efficacy by quantifying AtFH1-generated actin seeds over time with the different stoichiometries of XopR and formin (Fig. 4a). At an XopR-AtFH1 stoichiometry of 4:1, XopR displayed a robust promotion of AtFH1 activities in actin nucleation (Fig. 4b, c and Supplementary Fig. 6e, f). Surprisingly, at a high XopR-AtFH1 stoichiometry of 16:1, we

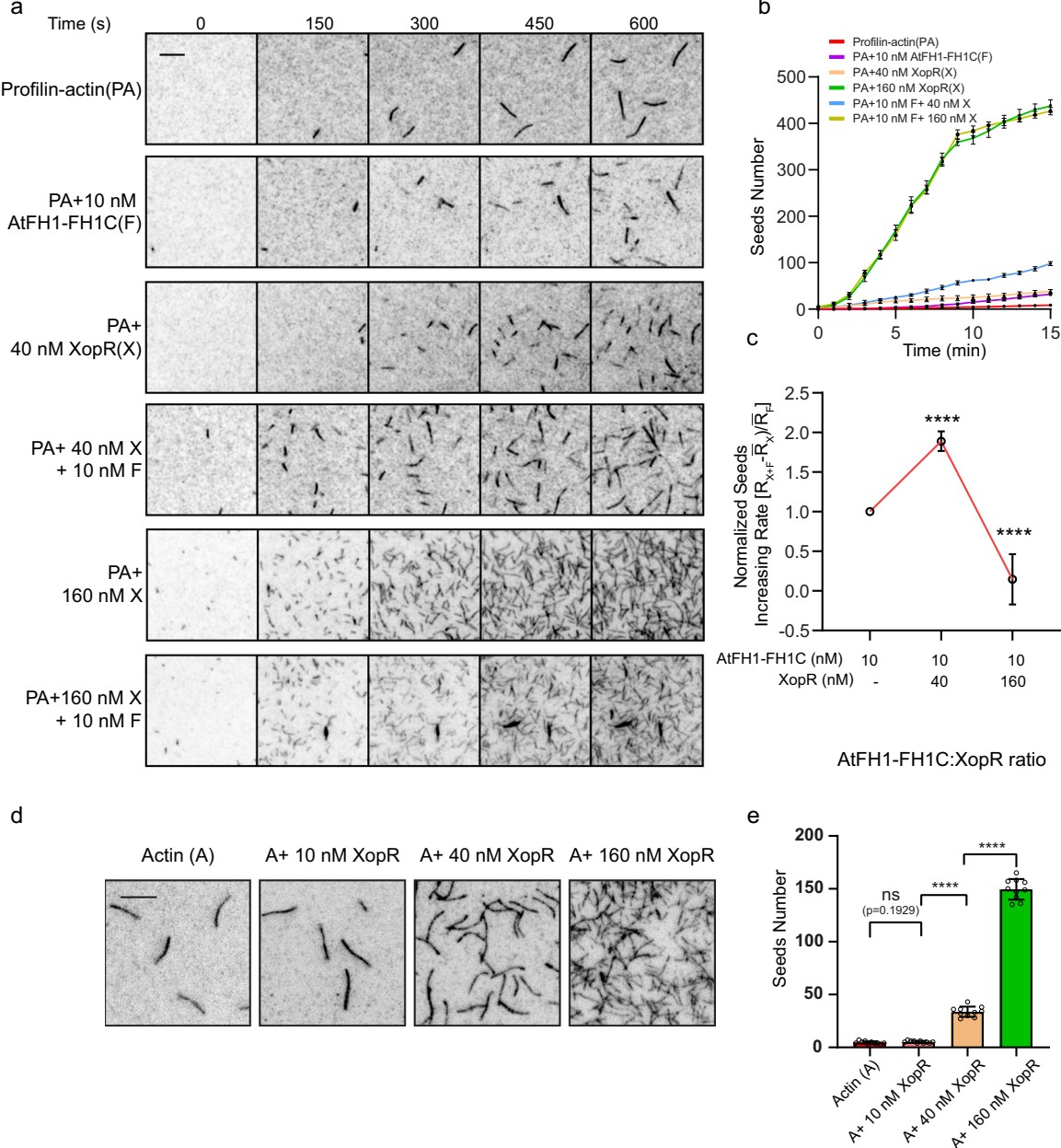

**Fig. 4 XopR modulates AtFH1-mediated actin nucleation in the TIRF-actin assembly assay. a** Representative time-lapse images from TIRFM where 0.5 μM G-actin (10% Oregon-actin) was incubated with 2 μM AtPRF1 and polymerized in the presence of the indicated concentrations of AtFH1-FH1C and XopR. **b** Actin seed number quantification in the area of $22 \times 22\ \mu m^2$ from **a** ($n = 6$, Error bar, SD). **c** Normalized actin seed increasing rate calculated from **b** ($n = 6$, Error bar, SD). Scale bar $= 5\ \mu m$ for **a**. **d** Representative images from TIRFM experiments where 0.5 μM G-actin (10% Oregon-actin) was polymerized in the presence of the indicated concentration of XopR for 300 s. **e** Quantification of actin seed number in the area of $22 \times 22\ \mu m^2$ from **d** ($n = 10$, Error bar, SD). Scale bar: 5 μm for **a** and **d**. Two-tailed Student's $t$-test was performed assuming equal variance. Ns no significant difference, $*p < 0.05$, $**p < 0.01$, $***p < 0.001$, $****p < 0.0001$.

observed a combined consequence of actin nucleation. First, in the tens of nanomolar range, XopR promoted AtFH1-mediated actin nucleation and also exhibited mild activity in triggering seed generation directly on its own (Fig. 4a–e). Second, at a high concentration of XopR (160 nM), a formin-mediated nucleation effect could not be observed, showing an inhibitory effect on AtFH1 activity at a 16:1 molar ratio of XopR:AtFH1, although 160 nM XopR demonstrated higher production of actin seeds on its own than 40 nM XopR (Fig. 4d, e). Furthermore, with an excess of G-actin at a concentration of 1.5 μM, formin activity was still abolished by XopR at the 16:1 stoichiometry of XopR: AtFH1 (Supplementary Fig. 6g–i). The above results

demonstrated a biphasic regulation of AtFH1 activities by XopR over a range of XopR:AtFH1 stoichiometries at a level of several tens of nanomoles, where XopR alone also demonstrated actin nucleation activities.

**XopR binds actin via a WH2-like domain and bundle F-actin through increasing multivalency.** We hypothesize that the nucleation activities of XopR might occur because of its self-association (Fig. 2c) if XopR can interact with G-actin directly. Indeed, sequence analysis of XopR identified a WH2-like motif, although it is slightly different from the conventional WH2 motif

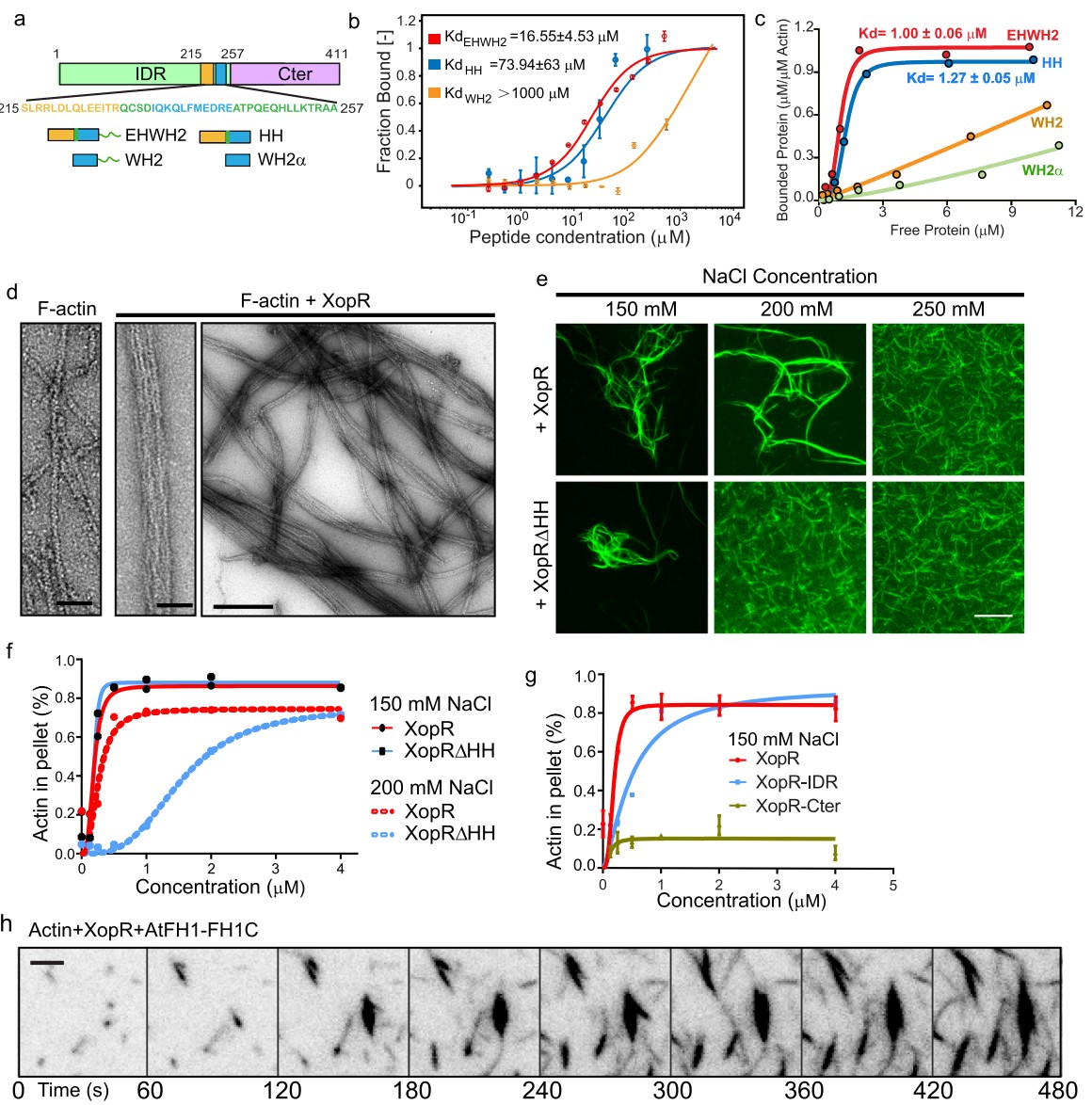

**Fig. 5 Binding and bundling of F-actin by XopR. a** Schematic domain illustration of four XopR truncation variants, EHWH2, HH, WH2, and WH2α. **b** MST binding curves of LatB-G-actin titrated with different XopR peptides with three biological replicates each. $n = 3$; Error bar, SD. **c** High-speed F-actin cosedimentation assay using MBP-EHWH2-msfGFP, MBP-HH-msfGFP, MBP-WH2-msfGFP, and MBP-WH2α-msfGFP. The data were fit using a Hill equation. **d** Negative stain electron microscopy (EM) of F-actin bundles formed by mixing 1 μM XopR with 0.2 μM F-actin in 150 mM NaCl. Scale bar from left to right: 400, 50, and 50 nm. **e** Micrographs of 0.2 μM F-actin in the presence of 10 μM XopR and XopRΔHH in the indicated NaCl buffer. F-actin was labeled with Acti-stain™ phalloidin. Scale bar = 5 μm. **f** Low-speed F-actin cosedimentation assay with XopR and XopRΔHH in the buffer with both 150 and 200 mM NaCl. **g** Low-speed cosedimentation assay of XopR full-length, XopR-IDR and XopR-Cter in 150 mM NaCl solution ($n = 3$ biological replicates). Data were presented as mean values ± SD. The data were fit using a Hill equation. **h** Representative time-lapse images of TIRF-actin polymerization over 480 s with 0.5 μM G-actin (10% Oregon-actin), 100 nM AtFH1-FH1C, and 400 nM XopR under 50 mM NaCl conditions. Scale bar = 2 μm for **f**.

in having an extended α-helix (EH) (SLRRLDLQLEEITR) at the N-terminus of the WH2 α-helix (IQKQLFMEDRE) (Fig. 5a and Supplementary Figs. 2d, 6j). We then investigated the interactions of WH2-like motifs of XopR with G- and F-actin. We synthesized four peptides with different combinations of EH (Fig. 5a). Using a label-free microscale thermophoresis (MST) assay, EHWH2, HH, and WH2 peptides were revealed to interact with G-actin directly at different Kd: EHWH2 = 16.5 ± 4.53 μM, HH = 73.94 ± 63 μM, and WH2 >100 μM, whereas no detectable binding for WH2α peptide was observed (Fig. 5b and Supplementary Fig. 6k). We next performed a high-speed cosedimentation assay to determine F-actin binding by four XopR peptide variants that were fused with an N-terminal MBP tag and a C-terminal msfGFP tag

(MBP-EHWH2-msfGFP, MBP-HH-msfGFP, MBP-WH2-msfGFP, and MBP-WH2α-msfGFP), which enabled the examination of short peptides binding to F-actin via gel electrophoresis. EHWH2 and HH showed similar F-actin affinity (Kd: EHWH2, 1.00 ± 0.06 μM; HH, 1.27 ± 0.05 μM), whereas the conserved WH2-like motif alone exhibited a much lower affinity to F-actin (Fig. 5c and Supplementary Fig. 6l).

Given the multivalent nature of XopR and its F-actin binding, we hypothesize that XopR may also crosslink F-actin to contribute to bacterial-induced actin bundling (Fig. 1c, e). We tested this hypothesis at physiologically relevant ionic strength with 150 mM NaCl using phalloidin-staining and low-speed sedimentation assays, in which XopR exhibited a prominent effect

in bundling F-actin (Fig. 5d, e), which depends on IDR but not the C-terminus of XopR (Fig. 5g and Supplementary Fig. 6m, n). Removal of the actin-binding motif of XopR (XopRΔHH) did not eliminate the bundling ability completely but resulted in higher sensitivity to salt perturbation. Full-length XopR, but not XopRΔHH, was able to maintain actin bundling starting from submicromolar concentrations with an increase in NaCl to 200 mM (Fig. 5e, f and Supplementary Fig. 6o, p). However, if we further increased the ionic strength to 250 mM, F-actin bundling by XopR was entirely abolished (Fig. 5e). The above results suggest that F-actin bundling by XopR is achieved by a joint effort of the HH motif and electrostatic-dependent interactions, likely the interaction between the positively charged surface patches of XopR and F-actin, which is a negatively charged polyelectrolyte. In addition, we asked whether, under a low ionic strength with a higher valency, XopR could drive actin to form actin-based liquid bundles, as previously shown by bundling-factor filamin[40]. Interestingly, XopR induced similar F-actin droplet formation in a tactoid shape from formin-nucleated short actin filaments within 50 mM NaCl (Fig. 5h and Supplementary Movie 3).

**XopR inhibits actin depolymerization factor (ADF)-mediated actin depolymerization.** Because the WH2 domain interacts with G-actin at a cleft between subdomains 1 and 3 that is also targeted by multiple binding partners, such as marine toxins, jaspinsamide, cofilin, and gelsolin (Supplementary Fig. 7a)[41], we hypothesized that XopR might compete with ADF/cofilin for F-actin depolymerization via steric effects and thereby contribute to the stabilization of bacteria-triggered actin bundles (Fig. 1c, e). We used TIRF- and pyrene-based F-actin depolymerization assays to examine AtADF3-mediated depolymerization[42] with and without the full-length, truncated variants, and HH peptides of XopR. XopR displayed potent inhibition of AtADF3, which depends on XopR-IDR, specifically the HH motif (Fig. 6a, b and Supplementary Fig. 7b–i, k, l). We also ruled out the possibility that XopR impairs depolymerization by manipulating AtADF3 directly because no evidence of interaction between XopR and AtADF3 was detected (Supplementary Fig. 7j). Furthermore, we also studied XopR-inhibited actin depolymerization in vivo by applying LatB to Arabidopsis expressing Lifeact-Venus. *Xcc*-infected seedlings showed more resistance to LatB treatment than *XccΔXopR*-infected seedlings at 24 hpi. In addition, XopR-overexpressing Arabidopsis showed apparent resistance to LatB-mediated actin depolymerization (Fig. 6c–e), which was consistent with XopR's role in inhibiting Arabidopsis ADF.

## Discussion

**IDR-mediated multifaceted and multivalent interactions for phytobacterial effectors.** T3Es selectively target host cellular hubs to manipulate various functional interactomes at the systems-level[43,44]. T3E-induced macromolecular assembly enhances the connectivity of T3E-host networks that impair the host resilience mechanism, which is attributed to intracellular scale-free biomolecular networks[45]. In addition, T3Es have coevolved adaptive mechanisms to evade surveillance from the plant ETI system, such as diversifying the sequence to avoid memorable patterns and suppressing various plant PTI pathways[6,46,47]. Phytobacterial pathogens preserved a large portion of intrinsically disordered residues in the evolution of T3Es[1], which introduced sophisticated invasion strategies by retaining high sequence diversity and interactive patterns for recognizing host biomolecules. Due to the flexible conformation and interactions of IDR, common molecular mechanisms underpinning IDR-containing T3Es for host invasion have not been identified[48].

Here, we investigated an IDR-containing T3E XopR that assembles host biomolecules gradually into a macromolecular complex via molecular condensation, which thereby dynamically manipulates the corresponding host pathway. We also found that bacteria subvert the plant system efficiently by developing the T3E that associates with the host surface scaffolding system, the PM-AC continuum, which generates local complex interactive networks in providing a focused attack. Our work explored a detailed mechanism that underlies the network interplay between pathogens and the front defense line of the plant host. The actin filaments and surface lipids of the PM-AC continuum cooperatively regulate the PM compartmentalization and lateral dynamics of plant surface molecules, many of which play essential roles in immune activation and signal transduction[13,49–52]. Subversion or hijacking of the host AC by bacterial pathogens is a well-known strategy for manipulating host cellular pathways or taking advantage of host materials during invasion[53,54]. In response, the host also rapidly reorganizes actin filaments upon the perception of virulence factors[55–57]. With multiple ABPs involved in orchestrating actin treadmilling, perturbations in a few ABPs were found to interfere with the plant actin remodeling during defense responses[58,59]. However, the mechanism of initiating actin remodeling during immune responses remains elusive. Recently, LLPS has been found to play essential roles in activating actin nucleation during T-cell signaling, in which the IDR and interactive motifs of multiple components of the protein complex cluster condense actin nucleation factors and enhance actin nucleation at the PM via multivalent interactions[14,15,60].

Our studies revealed a unique mechanism for host–pathogen communication, in which phytobacterial T3E subverts the AC on the PM through complex coacervation of multiple constituents of the PM-AC continuum. *Xcc* XopR targets multiple Arabidopsis surface biomolecules using an amphiphilic motif for its association with the PM inner leaflet, electrostatic attraction for interactions with the membrane surface, and F-actin, as well as direct binding with the juxtamembrane region of type I formins for formin clustering. *Xcc* gradually translocates T3E XopR into the host Arabidopsis, in which the AC is remodeled by XopR at multiple steps of actin treadmilling, including formin-mediated nucleation, G-actin binding, F-actin crosslinking, and depolymerization. The effectiveness of T3E-mediated subversion depends on the assembly of macromolecular condensates that undergo nanoscale two-dimensional LLPS.

**Biphasic regulation of formin activity by XopR via surface condensation at the nanocluster scale.** Acceleration of actin nucleation and processive elongation were also found for both Arp2/3-WASP nucleation complex[15,61] and Ena/VASP family nucleator[60,62], by increasing the valency and stoichiometry of the G-actin-binding sites. N-WASP-Arp2/3 assembly activated actin polymerization is positively correlated with the dwell time of N-WASP and Nck-N-WASAP stoichiometry, but not with the N-WASP density or monotonic concentration increase of Nck[15].

The regulatory mechanism that activates PM-inserted type-I formin in plants is unknown and is different from GTPase-mediated activation in mammals. Here we demonstrated biphasic regulation of plant formin activities in actin polymerization during bacterial infection in a formin-XopR stoichiometry-dependent manner (Fig. 7). Type I formin activity is activated during initial bacterial infection and XopR injection but inhibited at the late stage by more secreted XopR. Surface formins were condensed and their activity gradually finetuned by the incremental secretion of XopR in vivo during bacterial infection. At

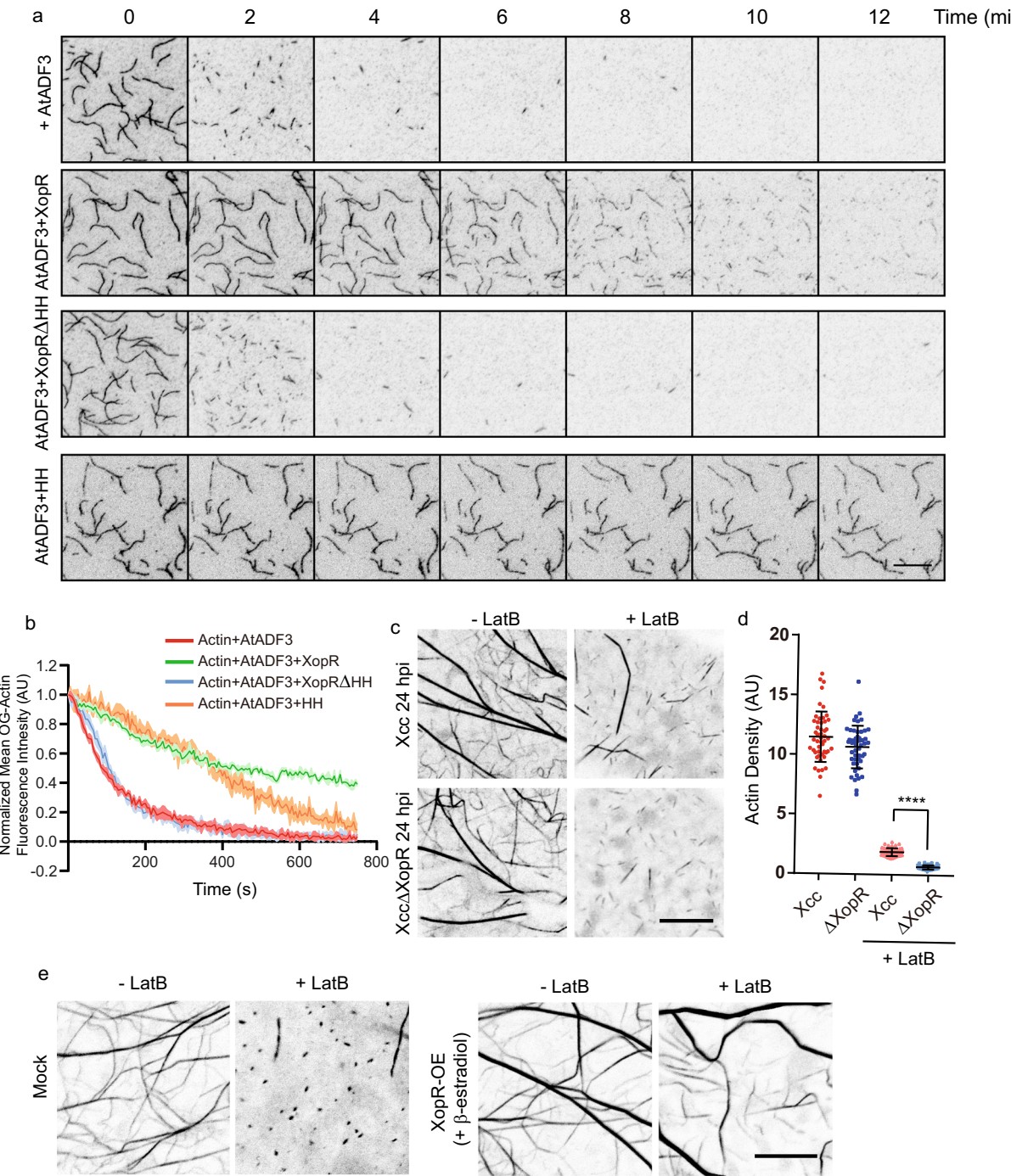

**Fig. 6 XopR inhibits ADF-mediated actin depolymerization through direct competition. a** Representative time-lapse TIRF images of F-actin depolymerization. To obtain these images, 100 nM XopR, 100 nM XopRΔCC, 0.2 μM AtADF3, and 20 μM CC peptide were used. **b** Mean fluorescence intensity of F-actin of **a** (n = 6, data were presented as mean values with error bands which represent SD). **c** Representative images of Lifeact-Venus in epidermal cells of WT *Arabidopsis* cotyledons. Seven-day-old seedlings were flood-inoculated with *Xcc* or *XccΔxopR*, and then treated with 1 μM actin LatB for 1 h after 24 hpi. Scale bar = 10 μm. **d** F-actin density quantification in **c** (n = 50 images from five individual seedlings (Data were presented as mean values ± SD). **e** Representative images of Lifeact-Venus in epidermal cells of *Arabidopsis* cotyledons expressing XVE-XopR. The expression of XopR was first induced for 24 h using 10 μM β−estradiol before being subjected to 1 μM LatB treatment for 1 h before imaging. Scale bar = 5 μm for **a**, Scale bar = 10 μm for **c**, **e**. Two-tailed Student's *t*-test was performed assuming equal variance. Ns no significant difference, *p < 0.05, **p < 0.01, ***p < 0.001, ****p < 0.0001.

the early stage with a low dose of XopR secretion, XopR bound rapidly to PM-integrated type I formin and concentrated formins into nanoclusters with low-level oligomerization. Concurrently, promoted plant actin nucleation was observed after XopR injection. Such an in vivo enhancement in actin nucleation could be the effect of two combined biochemical reactions. One is

XopR-mediated formin nanoclustering, which increases the avidity of formin, and thereby formin-mediated nucleation by creating a higher concentration of associated G-actin for processive barbed-end assembly. Another reaction that could occur concurrently is the enhanced spontaneous polymerization from XopR-G-actin interactions, which creates a

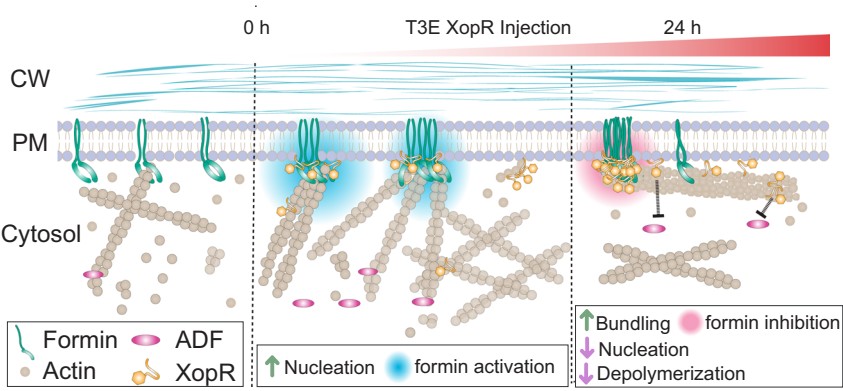

**Fig. 7 Remodeling of *Arabidopsis* actin by XopR during *Xcc* infection.** Schematic illustration of the molecular mechanisms by which bacterial effector XopR progressively hijacks plant host actin assembly. Using a type III secretion system, XopR stepwise manipulates different steps of host actin assembly, including activating formin-mediated nucleation at the early stage of infection and then stabilizing actin cytoskeleton at the later stage by crosslinking of F-actin and inhibiting actin depolymerization with a high protein level of XopR in the host.

thermodynamically favorable spontaneous assembly by recruiting multiple monomeric actins at the initial stage of infection. At the late infection stage with high amount of accumulated XopR (e.g., 24 hpi), XopR-mediated formin activation was converted to formin inhibition by the transformation of formin nanoclusters into condensed foci with higher-level clustering, confined movement, and inhibited activity in formin-mediated nucleation (Fig. 3a–e). We also recapitulated formin condensation in vitro using the SLB system, which showed a similar dose- and XopR-formin stoichiometry-dependent clustering of formin and changes in its nucleation activities in vivo (Figs. 3g, 4 and Supplementary Fig. 5a–g). A 4:1 stoichiometry of XopR:formin activated formin, and a 16:1 stoichiometry started to inhibit formin-mediated actin nucleation (Fig. 4). The biphase regulation of formin activity might indicate a change in the material properties and the increasing order of oligomerization for the multivalent formin-XopR complex over the increase of XopR dose and XopR-formin stoichiometry. At low XopR:formin stoichiometry during the initial XopR translocation, protein connectivity was increased and enhanced local cooperativity for actin nucleation, which might benefit from a local concentration increase of profilin-actin. As XopR continued to increase to the threshold, the XopR-formin complex at lower valency condensed into phase-separated complex coacervates with higher-order multivalency. Under such conditions, formin-mediated actin nucleation was significantly impaired both in vivo (Fig. 1f) and in vitro (Figs. 1f, 3i and 4a–c), possibly due to the high constraint on the flexibility of FH domains for processive barbed-end nucleation. The effective stoichiometry for formin inhibition could be lower in vivo, considering the highly crowded environment with more biomolecules on the 2D PM surface than under in vitro conditions with minimum components. Indeed, the transition and condensation of XopR-formin seem to be more abrupt in vivo than in vitro, because amorphous assemblies of the XopR-formin complex were observed when XopR was expressed at a high dose, which shaped the coacervates differently from in vitro spherical droplets on SLB. This result indicates additional partitioning of the unknown biomolecules in vivo that create adhesion or retention forces during complex coacervation and against the surface tension of the round droplets. Another contributing factor for differences in macro-molecular assembly kinetics observed in vivo on the membrane and in vitro in solution is the consideration of dimensionality. The 2D association allows a lower critical concentration of XopR to accelerate coacervate formation than that required in solution (Fig. 2e and Supplementary Fig. 5k). It is reminiscent of the plant

formin inhibition by the high-oligomerization of the actin-AtFH1-AtPRF3 complex during plant innate immune responses[11].

In contrast to other well-reported LLPS systems, such as RNA granules and pyrenoids[63,64], in which the physiologically relevant functions are closely associated with the large droplets at the micrometer and submicrometer scales, XopR starts to hijack and finetune formin activities at the nanometer scale with nanomolar concentration, which occurs during the formation of the nuclei at the early stage of LLPS (Figs. 3, 4, and Supplementary Figs. 4–6). Notably, in the case of LLPS in the nanomolar range, quantitative characterization of the phase boundaries of LLPS nucleation is technically challenging. Due to limitations in microscopic sensitivity and resolution, as well as time-resolution, in detecting the LLPS droplet formation, the commonly used approaches to defining the threshold concentration for a phase transition are usually applicable at the micromolar concentration, such as the bright spheres of XopR droplets shown in Fig. 2d, e, and Supplementary Fig. 5k. Most of the practical time points for imaging demixed droplets are in the range of minutes or hours, where the coacervates might not reach true equilibrium and thermodynamic coexistence, and therefore require a high concentration to detect the demixed droplets to define phase boundaries. The droplets using submicromolar or micromolar proteins supported the occurrence of the transition before the time point of imaging but were not accurate enough to quantitatively characterize LLPS nucleation. Here, we used single-particle TIRF imaging to resolve the threshold concentration of complex coacervation of XopR-formin around the nucleation of the phase transition by monitoring the dynamic fusion and clustering of individual formin puncta on the SLB (Fig. 3g and Supplementary Fig. 5f, g).

**XopR subverts multiple steps of actin assembly.** During the transition from a stimulatory stage of actin nucleation (early infection stage <12 hpi) to the inhibitory stage (later infection stage ~12–24 hpi), the accumulated XopR after 12 hpi starts to concurrently develop the second wave of actin attack through crosslinking and stabilizing F-actin that was previously polymerized by the XopR-formin complex. XopR binds F-actin via polyelectrolyte–counterion interactions and an evolutionarily distinct WH2-like domain. The multivalency of XopR further drives F-actin crosslinking, which depends on the biophysical properties and valency of the macromolecular complex of XopR and its binding partners, in an electrostatic-, stoichiometric-, and dose-dependent manner (Fig. 5). The XopR-mediated formin

clustering might also contribute to the actin-bundling. Recent multivalent formin also showed such an effect by generating crosslinked filaments[65]. Additionally, XopR directly competes with ADF for F-actin depolymerization, which leads to further stabilization of actin filaments in the plant. Nevertheless, during the same late stage (12–24 hpi), we also observed overall inhibition of F-actin production (Fig. 1c, d), which implied that the nucleation activity of XopR alone is also diminished in vivo. The inhibited XopR activity in actin nucleation might concurrently be derived from its slow increase in F-binding and bundling over time. Notably, XopR is also associated with the PM by its N-terminal amphipathic motif. Such cell surface association allows XopR proteins to target both surface formin and cortical actin filaments, which diluted their efforts in subverting any single molecular partner in the host but could provide a multifaceted influence on host biology (Fig. 7).

Phytopathogenic bacteria evolve and retain a large portion of IDRs in T3Es to facilitate flexible interactions with host biomolecules during attack and often attenuate host defense mechanisms[66]. The mechanisms underlying the remodeling of the PM-AC continuum and host signal transduction by T3Es are complicated due to the dynamic orchestration of a large set of ABPs in a spatiotemporally coordinated manner. Here, by integrating quantitative cell biology, biochemical, and biophysical studies, we have unraveled the complex mechanisms of phytobacterial T3E in hijacking plant actin assembly from the XopR family T3E. Via progressive coacervation at the cell cortex, XopR dynamically and spatiotemporally manipulates plant ABPs and actin during pathogenesis. The systematic dissection of the mechanisms by which XopR subverts the plant AC allowed us to better understand the complex host–bacteria interactions and may lead to the potential identification of the host guardee or decoy in addressing XopR-mediated Xcc infection. To date, however, the cortex-bound decoy system for bacteria–plant interactions remains unknown[4]. Our work will shed light on future challenges in elucidating LLPS and its functional consequences for pathogenic invasion and host defense mechanisms.

## Methods

**Arabidopsis thaliana**. The Columbia (Col-0) ecotype wild-type, 35 S:: GFP1-10[16], Lifeact-Venus line[18] with or without XVE:: XopR-mRuby2 as well as AtFH6-GFP line[32] with XVE:: XopR-mRuby2 were used in this study. Arabidopsis thaliana seeds were surface-sterilized using 15% bleach supplemented with 0.1% Tween-20 then plated onto 1/2 Murashige and Skoog (MS) medium supplemented with 0.8% sucrose, placed in the dark at 4 °C for 2 days then transfer to the growth chamber. Plants were grown at 22 °C under long-day conditions (16 h light/8 h dark cycles) unless otherwise stated. Generation of transgenic lines and phenotype analysis were described below.

**Bacterial strains**. Xanthomonas campestris pv. campestris (Xcc) WT, XccΔxopR, and the XccΔhrcC mutant were grown in NYG media (3 g/L yeast extract, 5 g/L peptone, 20 g/L glycerol, pH 7.0) at 30 °C and under agitation. The generation of the bacteria mutant was described below.

**Generation of transgenic Arabidopsis**. To create the transgenic XopR inducible expression line, XopR was constructed into a modified pER10 vector and mRuby2-FLAG was tagged at the C-terminal of XopR. Primers used in generating constructs were provided in Supplementary Table 1. Obtained constructs were transformed into both 35 S:: Lifeact-Venus and 35 S:: AtFH6-GFP Arabidopsis transgenic line using floral dip method via Agrobacterium strain GV3101 and screened by glufosinate-ammonium (45520; Sigma-Aldrich) on the ½ MS plate.

**Generation of bacteria mutant**. Xanthomonas campestris pv. campestris (Xcc) 8004 bacteria strain was used in this study. To generate XopR deletion mutant (XccΔxopR) and HrcC deletion mutant (XccΔhrcC), the upstream fragment and downstream fragment of the target gene were amplified and fused into plasmid pK18mobsacB. The plasmid was introduced into Xcc via the electroporation method. Positive colonies were screened on the NYG plate containing antibiotics. The deletion was further verified by colony PCR using target-specific primer pairs.

To generate the XopR-GFP11 strain, XopR-GFP11 fusion fragments were cloned into plasmid puj10 and transformed into XccΔxopR strain via the electro-transformation method. Positive colonies were screened by antibiotics and PCR. The primers and plasmids used are listed in Supplementary Tables 1 and 2.

**Bacterial infection**. Bacterial infection assay was done based on a modified protocol according to the previous report[67]. Briefly, seven-day-old Arabidopsis seedlings were chosen to do the Xcc flood-inoculation assay. Bacteria were harvested and resuspended into 40 mL of 0.02% Silwet L-77 (bioWORLD, USA), containing 10 mM MgCl$_2$, to a final concentration at OD = 0.1. The inoculum was dispensed into the 1/2 MS plate containing Arabidopsis seedlings, and the plates were incubated for 1 min at room temperature. Then the bacteria were removed by decantation, and the plates were sealed with 3 M Micropore Surgical Tape (3 M United States). The plates were incubated at 22 °C in the long-day growth chamber (16 h light/8 h dark). Images of bacteria-infected plants were taken at the indicated time points.

**Sequence analysis**. XopR sequence intrinsically disordered prediction was performed using the IUPRED2A algorithm (https://iupred2a.elte.hu/). IDR score larger than 0.5 is regarded as an intrinsically disordered region. XopR domain analysis was predicted using the conserved domain database (CDD) CD-search analysis (https://www.ncbi.nlm.nih.gov/Structure/cdd/cdd.shtml). Potential membrane-binding motif (MBM) was analyzed using an online tool AMPHIPA-SEEK (https://npsa-prabi.ibcp.fr/cgi-bin/npsa_automat.pl?page=/NPSA/npsa_amphipaseek.html). Coiled-coil domain was predicted using the COILS algorithm (https://embnet.vital-it.ch/software/COILS_form.html). The sequence alignment and phylogenetic tree analysis were performed by Clustal Omega (https://www.ebi.ac.uk/Tools/msa/clustalo/). The charge pattern of XopR was analyzed using the CIDER online server (http://pappulab.wustl.edu/CIDER/analysis/). Structure prediction of the EHWH2 region was performed using the PEP-FOLD3 algorithm (https://bioserv.rpbs.univ-paris-diderot.fr/services/PEP-FOLD3/).

**Protein reagents**. Xcc XopR-FL (aa 1–411), XopRΔHH (delete aa 215–243), XopR-IDR (aa 1–267), or XopR-Cter (aa 268–411) with a C-terminal His$_6$-Flag tag or a C-terminal mRuby2-His$_6$-Flag tag, Arabidopsis thaliana Formin 1 (AtFH1) AtFH1-FH1C (aa 430–1151) with an N-terminal GST-His$_6$ tag, Arabidopsis thaliana Formin 6 (AtFH6) AtFH6-FH1C (aa 293–899) or profilin 1 (AtPRF1) (aa 1–131) with an N-terminal His$_6$ tag, Arabidopsis thaliana actin depolymerization factor 3 (AtADF3) (aa 1–134) with a C-terminal His$_6$-Flag tag, Lifeact-msfGFP with an N-terminal His$_6$-MBP tag, chimeric protein EHWH2, HH, WH2, or WH2α with an N-terminal His$_6$-MBP and C-terminal msfGFP tag, Supercharged GFP(−30) were expressed and purified from E.coli BL21 (DE3) Rosetta. G-actin was purified from rabbit skeleton muscle acetone powder. Rhodamine-labeled and Pyrene-labeled actin were purchased from Cytoskeleton. ATTO488-actin was purchased from Hypermol. The used primers and generated constructs in this study are listed in Supplementary Tables 1 and 2.

**His-tag protein purification**. Protein expression plasmid was transformed into BL21 (DE3) Rosetta and selected by antibiotics. BL21 Rosetta cells were grown in 2 L of TB medium to an optical density of 1–2 at 600 nm, and expression was induced with isopropyl-thio-β-D-galactoside (IPTG, 0.5 mM) at 16 °C. Cells were harvested after overnight induction, washed and resuspended in 20 ml of Buffer A (20 mM HEPES, pH 7.4, 500 mM NaCl, 20 mM Imidazole, 10% Glycerol) and lysed with LM20 Microfluidizer™, and the clarified lysate was loaded onto a 5 mL HisTrap HP column (GE Healthcare) connected to an FPLC AKTAxpress system (GE Healthcare). The column was washed by 20 mM HEPES pH 7.4, 500 mM NaCl, 20 mM Imidazole (Buffer A) and eluted by 20 mM HEPES pH 7.4, 500 mM NaCl, and 500 mM Imidazole (Buffer B). The proteins were further purified by gel filtration on a HiLoad 16/600 Superdex 200 pg column (GE Healthcare) equilibrated with 20 mM HEPES pH 7.4, 300 mM NaCl, 10% Glycerol (GF Buffer), and concentrated in concentrators (Amicon Inc.) to ~10 mg/ml. Protein concentrations were determined using Nanodrop 2000.

**Actin purification**. Lyophilized rabbit skeleton muscle acetone powder was rehydrated and ground in G-buffer (5 mM Tris-HCl, pH 8, 0.2 mM ATP, 0.1 mM CaCl$_2$, 0.5 mM DTT, annd 0.1 mM sodium azide), and then cleared by centrifugation at $27,000 \times g$ in the JA25.50 rotor (Beckman Coulter, Inc., USA). Solubilized actin in the supernatant was collected and polymerized by adding 50 mM KCl and 2 mM MgCl$_2$ for 1 h, followed by the addition of 0.8 M KCl for 30 min at 4 °C. F-actin was pelleted by centrifugation at $147,600 \times g$ using a Type50.2Ti rotor (Beckman Coulter, Inc., USA) and then depolymerized by brief sonication and dialysis against G-buffer for 48 h at 4 °C. Monomeric Ca$^{2+}$-ATP-actin was cleaned by centrifugation at $193,900 \times g$ for 2.5 h using the SW 55Ti rotor (Beckman Coulter, Inc., USA). The supernatant was collected and loaded to Sephacryl S-300 HR for gel filtration chromatography using G-buffer (5 mM Tris-HCl, pH 8, 0.2 mM ATP, 0.1 mM CaCl$_2$, 0.5 mM DTT, and 0.1 mM sodium azide) to obtain monomeric Ca$^{2+}$-ATP-actin.

**Analytical gel filtration**. To determine the size of XopR in different concentrations of NaCl saline, analytical gel filtration analysis was carried out using a Superdex 200 Increase 10/300 column (GE Healthcare). The column was balanced using a different salt buffer for the size determination of XopR at different NaCl concentrations. Protein standards were fresh prepared containing 10 mg/mL albumin (66 kDa), 10 mg/mL apoferritin (443 kDa), 8 mg/mL thyroglobulin (669 kDa), 5 mg/mL alcohol dehydrogenase (150 kDa), 4 mg/mL β-amylase (200 kDa), and 3 mg/mL carbonic anhydrase (29 kDa). The standard mix was applied to the column using the same sample volume as the protein to be analyzed.

**Protein labeling**. The primary amine group of AtFH1-FH1C, AtFH6-FH1C, and XopR were labeled by Alexa Fluor$^{TM}$ 488, Alexa Fluor$^{TM}$ 561, and Alexa Fluor$^{TM}$ 647 labeling kit (Thermo Scientific). Briefly, the protein of interest was prepared at 2 mg/mL in buffer containing 0.1 M sodium bicarbonate buffer (pH 8.0–8.3). Alexa dye was added in and incubated with protein for 1 h at room temperature or overnight at 4 °C. Excess dye was removed from labeled protein by using HiTrap Desalting column (GE Healthcare). Labeling efficiency was determined using Nanodrop 2000.

Oregon-actin labeling was performed according to published paper[68]. Briefly, monomeric actin was dialyzed against G-buffer without DTT. Actin was polymerized by mixing an equal volume of cold 2x label buffer (50 mM imidazole, pH 7.5, 0.2 M KCl, 4 mM MgCl$_2$, 6 mM NaN$_3$, and 0.6 mM ATP). After 5 min, polymerized actin was diluted to 1 mg/ml (23.3 μM) with cold 1 × label buffer. A fresh 10 mM stock solution of Oregon green 488 iodoacetamide (Thermo scientific) was prepared in N, N-dimethylformamide. A 12- to 15-fold molar excess of Oregon green was added dropwise to the actin while gently vortexing, and the solution was rotated overnight at 4 °C avoid of light. Labeled actin was pelleted by centrifugation at 160,000 × g for 2 h. The pellet was resuspended and homogenized in bounce buffer (3 mM Tris pH 8.0, 10 mM DTT, 0.2 mM ATP, and 0.1 mM CaCl$_2$), sonicated for 1 min, and dialyzed for 2 days with G-buffer at 4 °C avoid of light. After centrifugation at 440,000 × g for 1 h, the supernatant was loaded onto Hiload Superdex 200 16/600 column, and Oregon-actin containing fractions were collected. The concentration of actin and Oregon green were determined based on OD$_{290}$ and OD$_{491}$ reading from Nanodrop 2000.

**Surface plasmon resonance (SPR)**. SPR experiment was performed by the Biacore T200 instrument version 2.0.1 (GE Healthcare) at room temperature in buffer containing 20 mM HEPES, 150 mM NaCl pH 7.4. Ligand-protein AtFH1-FH1C, AtFH6-FH1C, XopR, AtPRF1, and AtADF3 were immobilized on the CM5 chip (GE Healthcare) by amine coupling. The carboxyl group on the dextran surface of the chip was converted amine-reactive ester by reacting with 0.2 M 1-ethyl-3-(3-dimethylpropyl)-carbodiimide and 0.1 M N-hydroxysuccinimide. The ligands to be immobilized were then injected to the surface at a flow rate of 10 mL/min at pH 4.5, while the reference cell was left blank without injected protein. To test the binding of analytes with the immobilized, analyte was flown in over the surface of the control and ligand for 60 s. The dissociation between ligand and tested binding partners were disassociated with washing buffer (20 mM HEPES, 150 mM NaCl, pH 7.4) for 150 s at a rate of 30 μL/min. The concentration of XopR ranged from 11 to 0.043 μM. The chip surface with left-over protein captured on was regenerated by treating with 50 mM NaOH for 3 s at 100 μL/min after each cycle. The kinetics of binding was analyzed by Biacore T200 Evaluation software version 3.0 (GE Healthcare). The sensorgrams for the binding experiment were normalized with the reference cell and fitted to the bivalent analyte model.

**Pyrene–actin polymerization and depolymerization kinetics**. Pyrene fluorescence was monitored at excitation 365 nm and emission 407 nm at 25 °C in a fluorescence spectrophotometer BioTek Cytation 5 multimode imaging reader. For actin polymerization, 2 μM G-actin (3% pyrene-labeled actin) was mixed with protein to be tested and polymerization was initiated by adding 10X KME (500 mM KCl, 10 mM MgCl$_2$, and 10 mM EGTA). For actin depolymerization assay, 4 μM G-actin (30% pyrene-labeled actin) was preassembled for 2 h to get filamentous actin. Actin depolymerization was initiated by 40 times dilution using F-buffer either alone, or with protein to be tested. Rates of the assembly were calculated from the slopes of curves at 0–5 min.

**Immunoblotting**. XopR expression was induced by 10 μM β-estradiol treatment. Plant samples were collected at different induction times and were homogenized by grinding in liquid nitrogen. The plant powder was then resuspended in extraction buffer (500 mM Tris-HCl, 250 mM sucrose, 0.5 M EDTA, 5% glycerol, 50 mM KCl, 50 mM sodium pyrophosphate, 25 mM sodium fluoride, 1 mM sodium molybdate, 2 μg/mL chymostatin, 2 μg/mL aprotinin, 2 μg/mL pepstatin, and 1 mM PMSF). Proteins were detected using the primary antibodies FLAG M2 monoclonal antibody produced in mouse (1:2000; Sigma-Aldrich, Cat # F1804) and IRDye® 800 CW Goat anti-mouse secondary antibody (1:10000; LI-COR Biosciences, Cat # 926-32210). Blots were subsequently scanned using Odyssey Infrared Imager (LI-COR Biosciences).

**Agrobacterium-mediated transient expression in Nicotiana benthamiana**. Nicotiana benthamiana plants were grown at 24 °C in a growth chamber under long-day condition (16-h-light/8-h-dark cycle). A. tumefaciens strain GV3101 carrying plasmid harboring 35 S::AtFH6-GFP, XVE::XopR-mRuby2, and XVE:: XopR-Cter-mRuby2, respectively, were streaked on LB plate with 25 μg/ml Rifamycin and 50 μg/ml spectinomycin and incubated at 30 °C for 2 days. Pick single colony and culture overnight in LB solution with 25 μg/ml Rifamycin and 50 μg/ml spectinomycin. Bacterial cells were harvested and resuspended in induction media containing 10 mM MES (pH = 5.6), 10 mM MgCl$_2$, and 200 μM acetosyringone for 2 h at room temperature before inoculation. Five–six-week Nicotiana benthamiana leaves were press-infiltrated with agrobacteria at a concentration of OD$_{600}$ = 0.5 by 1 mL syringe at abaxial side. After injection, plants were put back to growth chamber. At 24 hpi (hours post infiltration), Nicotiana benthamiana leaves were spread with 20 μM β-estradiol. Images were taken at 48 hpi.

**Microscopy and image analysis**

*For actin images*. To quantify F-actin changes after pathogen infection, images were acquired by Zeiss Elyra PS.1 super-resolution system confocal mode using the Zeiss Alpha Plan Apochromat 100x, NA 1.46 oil objective. Seven-day-old Lifeact-Venus Arabidopsis actin reporter line was flood-inoculated with bacteria. The epidermal cells from the cotyledon region were chosen for image acquisition.

*For formin images*. To quantify AtFH6-GFP changes after XopR overexpression, images were acquired by the Zeiss Elyra PS.1 super-resolution system confocal mode using the Zeiss Alpha Plan Apochromat 63x, NA 1.46 oil objectives. Seven-day-old 35 S::AtFH6-GFP and XVE::XopR-mRuby2 transgenic lines were used for imaging. XopR overexpression was induced by 10 μM β-estradiol. To observe AtFH6-GFP pattern change after pathogen infection, seven-day-old AtFH6-GFP Arabidopsis formin reporter line was flood-inoculated with bacteria. The epidermal cells from the cotyledon region were used for imaging. Time-lapse images were captured by the sCMOS camera with 100 ms exposure time and no interval.

*For splitGFP images*. To observe real-time delivery of effector, seven-day-old 35 S:: GFP1-10 Arabidopsis line was flood-inoculated with Xcc XopR-GFP11 bacterial strain. Epidermal cells from the cotyledon region were chosen to do imaging. The image was carried out in the Zeiss Elyra PS.1 super-resolution system VA-TIRFM mode using the Zeiss Alpha Plan Apochromat 63x, NA 1.46 oil objective.

*For actin TIRF and reconstitution assays images*. For actin TIRF assays and SLB reconstitution assays, a Nikon Ti-E inverted microscope (Nikon, Shinagawa, Tokyo) was used. The microscope was equipped with a perfect focus system that prevents focus drift, an iLAS2 motorized TIRF illuminator (Roper Scientific, Evry Cedex, France), and Prime95b sCMOS camera (Photometrics, Tucson, AZ). All images were acquired using objective lenses from Nikon's CFI Apochromat TIRF Series (100xH N.A. 1.49 Oil). Multi-channel imaging of samples was achieved by the sequential excitation with 491 nm (100 mW), 561 nm (100 mW) and 642 nm (100 mW) lasers, reflected from a quad-bandpass dichroic mirror (Di01-R405/488/561/635, Semrock, Rochester, NY) located on a Ludl emission filter wheel (Carl Zeiss AG, Oberkochen, Germany). The microscope was controlled by MetaMorph software (Molecular Devices, LLC, Sunnyvale, CA).

*For fluorescence microscope images*. To monitor phase separation droplets and F-actin bundling induced by XopR and XopR truncating variants, samples were prepared and imaged on Leica DMi8 (Leica Microsystems, Germany) equipped with an HCX PL APO 100x/1.4 OIL objective, ORCA-Flash4.0 LT (Hamamatsu, Japan), and a solid-state Spectra-X light engine (Lumencor, USA). All the images were acquired using Metamorph software (Molecular Devices, USA) and processed using ImageJ.

**Fluorescence recovery after photobleaching (FRAP) of formin clusters on the SLB**. For in vivo AtFH6-GFP oligomerization state analysis, seedlings were transferred to 500 mL of 4% paraformaldehyde (P6148, Sigma-Aldrich) in PBS and fixed for 2 h at room temperature with gentle orbital shaking. A high laser power (100%) was used, and time-series images were acquired with an exposure time of 200 ms (no interval) to record the particle signal until all the particles were totally bleached. To induce the bleaching of AtFH1-GFP puncta on non-dynamic SLB (79.5% POPC, 20% DGS-NTA-Ni+, 0.5% Rhod PE), a high laser power (~50%) was used, and time-series images were acquired with an exposure time of 50 ms (no interval) to record the particle intensity until all the visible particles were bleached. To check SLB dynamics, FRAP experiment was performed with an Elyra PS.1 super-resolution system (Zeiss, Germany) confocal mode. The region of interest was bleached with a 100% power of 561 nm laser pulse. Recovery from the photobleaching area was recorded in a single focal plane.

**Microscale thermophoresis (MST)**. Binding affinity between monomer actin and the CCWH2/CC/WH2/WH2α peptides was measured by MST method (Duhr and Braun, 2006). Briefly, EHWH2 peptide (SLRRLDLQLEEITRQCSDIQKQLFME-DREATPQEQHLLKTRAA), HH peptide (SLRRLDLQLEEITRQCSDIQKQLF-MEDRE), WH2 peptide (QCSDIQKQLFMEDREATPQEQHLLKTRAA), and WH2α peptide(QCSDIQKQLFMEDRE) were synthesized by GL Biochem (China).

Binding reactions were carried out in buffer containing 50 mM Tris pH 7.4, 150 mM NaCl, 0.05% Tween 20. Peptides were serially diluted (1:1) and titrated into 10 μM LatB-Actin (1:1). Samples were loaded into Monolith™ NT.115 premium capillaries immediately after preparation to avoid unspecific adsorption. Before MST measurement, the reaction was incubated at 22 °C mounted in the Monolith™ NT.115 apparatus (Nanotemper Technologies). The data were collected at 22 °C using the red LED at 5% (GREEN filter; excitation 515–525 nm, emission 560–585 nm) and IR-Laser power at 40%. Data analyses were performed with NTAnalysis.

**Cosedimentation assay.** Actin filaments were assembled first for 2 h at 25 °C in F-buffer (10 mM Tris-HCl pH 7.5, 50 mM KCl, 2 mM MgCl$_2$, 1 mM EGTA, 0.2 mM DTT, 0.2 mM ATP, and 0.2 mM CaCl$_2$). A concentration of 2 μM actin filaments was incubated with a range of concentrations of the target protein for 20 min at 25 °C and then spun at 10,000 × g (low speed) or 100,000 × g (high speed) for 20 min at 25 °C. Equal volumes of the total, supernatant, and pellet samples were separated by SDS–PAGE, stained with GelCode Blue Safe protein stain.

**Reconstitution assays on the SLB.** Liposome was prepared as previously described from a mixture of 1-palmitoyl-2-oleoyl-glycero-3-phosphocholine (POPC), 1,2-dioleoyl-sn-glycero-3-[(N-(5-amino-1-carboxypentyl) iminodiacetic acid)succinyl] (nickel salt) (DGS-NTA-Ni$^+$) supplemented with 0.5% 1,2-dipalmitoyl-sn-glycero-3-phosphoethanolamine-N-(lissamine rhodamine B sulfonyl) (16:0 Liss Rhod PE)[69,70]. The lipid mixture in chloroform was evaporated under nitrogen gas and further dried under vacuum for 2 h. The mixture was rehydrated with PBS (pH 7.4), sonicated for 30 min using a water bath sonicator, followed by freeze-and-thaw cycles between −200 and 42 °C for 20 times, and then centrifuged for 45 min at 35,000 × g. SLBs were formed freshly in a 96-well glass-bottom plate precleaned by Hellmanex™ III and NaOH. Fifty microliters of liposome solution containing 0.5–1 mg/mL lipid were added to the coverslips and incubated for 30 min. Unabsorbed vesicles were removed by washing with basic buffer (50 mM HEPES, 150 mM NaCl, and 1 mM TCEP) extensively, and bilayers were blocked by washing three times with reaction buffer (50 mM HEPES, 150 mM NaCl, 1 mM TCEP, 1 mg/mL BSA, pH 7.4), and incubating for 20 min. Proteins were added into the chamber for 5 min then wash away by reaction buffer. To generate dynamic and non-dynamic SLB for TIRF imaging, 2% or 20% of DGS-NTA-Ni$^+$ was added accordingly to prepare SUV and form SLB. For in vitro formin size and bleaching step analysis, non-dynamic SLB (79.5% POPC, 20% DGS-NTA-Ni+, and 0.5% Rhod PE) was prepared based on the description above, and for formin fusion and XopR 2D-phase separation on the SLB, dynamic SLB (97.5% POPC, 2% DGS-NTA-Ni+, and 0.5% Rhod PE) was prepared for protein conjugation and imaging. Microscopy experiments were performed in the presence of a glucose/glucose oxidase/catalase O$_2$-scavenging system.

**Negative-stain transmission electron microscopy.** To check filaments bundling, 5 μM monomeric Ca$^{2+}$-ATP-actin was pre-polymerized in F-buffer for 1 h, then 4 μL of 5 μM filaments were incubated together with final 1 μM of XopR proteins in F-buffer in total 10 μL volume at room temperature for 10 min. Copper grids were purchased from Electron Microscopy Sciences and hydrophilized with a tabletop plasma cleaner for 30 s of UV light exposure, then 4 μL of the mixed sample was applied to the gird and waited for 2 min. Extra sample was removed with filter paper, and the grid was negatively-stained with 2% uranyl acetate for 1 min. Grids were imaged with an FEI tecnai T12 transmission electron microscope operating at 120 kV equipped with an Ultrascan 1000 CCD camera (Gatan, Inc.).

**Surface force apparatus (SFA) measurement.** In all SFA experiments, ruby mica, grade 1 (V-1/V-2), was obtained and used from S and J Trading, Inc. This mica was cleaved into smaller step-free, molecularly smooth pieces. Silver (99.99% pure) was purchased from Cerac Incorporated and deposited onto the backside of this freshly cleaved mica by thermal evaporation (Kurt Lesker nano36), resulting in a metal thickness of ~55 nm. The silvered mica was then cut into small rectangular pieces of ~1 cm$^2$ area, and glued silver side down onto curved cylindrical silica disks (ca. 2 cm radius of curvature). The glue was EPON Resin 1004 F from Momentive Specialty Chemicals Inc. More details of experiments can be found in the previous study[71]. The interaction forces between two mica surfaces in different solutions and the interfacial energy of the coacervate were measured using an SFA (SurForce LLC). The detailed experimental setup of the SFA has been reported elsewhere[72]. Briefly, thin silver-backed mica sheets (~5 μm thickness) were glued onto cylindrical silica disks (radius $R = 1$ cm). Twenty microliters of the coacervate solution was injected between two mica surfaces and was equilibrated for at least 20 min. The interaction forces and the separation distance between the surfaces were determined in situ and in real-time, using multiple-beam interferometry. The approach and separation speed of our SFA measurements was about 5 nm/s. The measured pulling force, $F_{ad}$, is correlated to the EIE $\gamma_{eff}$ by $\gamma_{eff} = -F_{ad}/4\Pi R$ (1)[27]. The unit of $\gamma_{eff}$ is mN/m which is equivalent to mJ/m$^2$. The concentration of XopR, AtFH1-FH1C, and ScGFP(−30), which are dissolved in HEPES buffer, are 10, 5, and 10 μM separately. The concentration of the NaCl is 50 mM unless otherwise specified.

**Imaging quantification**

*For actin density and bundling analysis.* Region of interest was cropped and converted to an 8-bit grayscale image. The percentage of occupancy (density) and skewness (bundling) were analyzed using Skewness and Density JAVA scripts in ImageJ.

*For mean square displacement (MSD) analysis.* Region of interest was cropped and subjected to particle tracking in SpatTrack. Images were de-noised for better particle detection. The particle detection threshold for each image was determined by the "find threshold" function in SpatTrack. The particle trajectories were generated in SpatTrack and MSD was calculated based on the equation listed below:

$$\text{MSD}(t) = \frac{1}{N-n} \sum_{i=1}^{N-n} \left\{ (x(i+n) - x(i))^2 + (y(i+n) - y(i))^2 \right\} \quad (2)$$

Where x and y indicate the particle position, the $N$ is the total frame number of the trajectory, $n$ is the frame number corresponding to $t$. Subsequently, to simply indicate the particle dynamics, the diffusion coefficient ($D_{eff}$) was calculated by fitting the MSD for each trajectory with browning diffusion analytical model: MSD $(t) = 4Dt$ (3), in which D is the $D_{eff}$.

*For bleaching step counting.* To count the bleaching step of AtFH6-GFP particles, the time-lapse images were analyzed in the Trackmate plugin in ImageJ (FIJI) to do particle tracking. The estimated particle diameter was set to 0.35 μm for particle detection. The final tracked particles were manually checked to ensure full particle bleaching. Then the outputted particle intensity was used to bleaching step counting.

**Statistics and reproducibility.** All experiments are performed with at least three biological replicates unless specially stated in the figure legend. All statistical analyses were performed using GraphPad Prism software (GraphPad, San Diego, CA, USA). P values were determined by two-tailed Student's t-test assuming equal variances (*$p < 0.05$, **$p < 0.01$, -***$p < 0.001$, ****$p < 0.0001$, and ns no significance). Data are reported as the mean ± SD or ±SEM, which is specified in the figure legend.

**Reporting Summary.** Further information on research design is available in the Nature Research Reporting Summary linked to this article.

## Data availability
Other data that support the findings of this study are available from the corresponding authors on reasonable request. Source data are provided with this paper.

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

## Acknowledgements

We thank Min Wu (NUS, Singapore) for sharing the Nikon imaging system for TIRF microscopy. We thank Atul Parikh (UC Davis) for the discussion on the design of biophysical experiments. We thank Ms Casandra Ai Zhu TAN from Liang Yang Lab (SCELSE, NTU, Singapore) for helping in generating the XopR deletion strain. We thank Ms Tingyan Dong (South China Agriculture University, China) for generating XopR-GFP11 and *XccΔhrcC* stain. We thank Gitta Coaker (UC Davis) for sharing *Arabidopsis thaliana* plants expressing GFP1-10. We also thank the NTU Protein Production Platform (www.proteins.sg) for the initial protein expression test and purification of XopR. This study was supported by NTU startup grant (M4081533), MOE Tier 1 (RG32/20), and MOE Tier 3 (MOE2019-T3-1-012) to Y.M. in Singapore.

## Author contributions

H.S., X.Z., C.L., Z.M., X.H., and Y.L. conducted the experiments. L.Y., J.Y., and Y.M. designed the experiments and analyzed the data. H.S. and Y.M. wrote the manuscript, which all authors edited.

## Competing interests

The authors declare no competing interests.
