## [Peer Review File · Nature Communications]

REVIEWER COMMENTS

Reviewer #1 (Remarks to the Author):

Interactions between intrinsically disordered regions (IDRs) are shown to drive phase separation for many proteins. Sun et. al. show that IDR-containing phyto bacterial type III effector (T3E) XopR hijacks the Arabidopsis actin cytoskeleton actin network in vivo by driving formin clustering and inhibiting depolymerization. The authors report that XopR IDR drives XopR phase separation and mediates actin polymerization and depolymerization in vitro by interacting with formin, G-actin and F-actin. This work provides mechanistic insights on how T3E IDR help avoid host defense via phase separation. The most exciting discovery, to me, is that 'by tuning the physical-chemical properties of XopR-complex coacervates, XopR progressively manipulates multiple steps of actin assembly, including formin-mediated nucleation, crosslinking of F-actin, and actin depolymerization by competing for actin-depolymerizing factor, depending on constituent stoichiometry.' My slight disagreement would be what physical-chemical properties are turned and how. The authors provided a biphasic model for the regulation of formin activity by XopR where at early stage low concentration of XopR clusters formin to promote actin nucleation and at later stage high concentration of XopR inhibits formin by 'transforming the formin nanoclusters into condensed foci with higher-level clustering'. The main evidence is that at high XopR concentration, actin seed number with or without formin is the same (Fig 4B). The authors reason that 'the biphasic regulation of formin activity might indicate a change in the material properties from the gelation without phase transition to gelation plus phase separation'. It is not clear whether the critical concentration of phase separation is the same at which inhibition for formin activity starts. Given the curves in Fig 4B start to have a smaller slope at about 10mins at 160 nM XopR, I suspect other parameters are limiting the nucleation, maybe the amount of available G-actin. If enough free actin is available for both formin and XopR, XopR+formin may still be more efficient at nucleating actin than XopR alone at high XopR concentration. The author can use higher G-actin at high XopR concentration to rule out this possibility. Even formin is inhibited, the seed number is still dramatically increased. This does not support XopR phase separation inhibits actin polymerization. XopR indeed directly nucleates actin, efficiently. An alternative interpretation is that high concentration of XopR is so effective at nucleating actin, the role of formin is not important anymore. Inhibition of polymerization at high XopR concentration could be attributed to binding of XopR to the polymerized actin. The authors focus on the effect of XopR concentration change overtime, another parameter in the system that is changing overtime is the actin: from monomer, to polymer, to bindles. I see a continuum of phase separation from XopR with formin to initiate actin nucleation, to XopR with G actin to further promote actin polymerization, to XopR with F actin to form bindles, which then inhibits actin depolymerization by forming a liquid to coat the actin and prevent access of depolymerization factors. In this multicomponent phase separation system, IDR in XopR provides the weak transient interaction needed for phase separation, formin dimerization and location on membrane (reduced mobility) decrease concentration needed for XopR phase separation, which is further reduced by the nucleated actin polymer. Supporting this model, polymerization promotes phase separation by increasing valence and actin binding proteins form liquid coat on actin bindles have been reported (Sanders, Cell 2020 P306-324.E28., Weirich, PNAS 2017 114 (9) 2131-2136). ED Fig. 6K clearly show XopR coating on actin bindles and I even see larger foci in XopR channel that are not present in the actin channel. Can the authors follow the kinetics of actin nucleation to polymerization and to bindle in the system while observing XopR phase separation? That would help differentiate the two models. If not, make sure there is excess G-actin for XopR in nucleation and compare the concentrations at which XopR phase separation happens and formin inhibition starts

would be informative (on the same SLB). Other minor concerns are:

1. I don't see deletion of XopR IDR on effect of formin clustering or actin polymerization/depolymerization in cells, which will provide direct evidence for the functional role of IDR phase separation, if doable.
2. I don't understand why the authors inhibit SLB fluidity. That would enhance protein phase separation. Is the PM less dynamic than SLB?
3. 'Here, we used single-particle TIRF imaging to resolve the threshold concentration of complex coacervation of XopR-formin around the nucleation of phase transition by monitoring the dynamic fusion and clustering of individual formin puncta on the cell surface (Fig 3g and Extended Data Fig 5f)'. My understanding is 3g are from Extended Data 5G, which are immobilized SLB. While 5f is in mobile SLB and there is no XopR concentration noted. They are not on the cell surface. I don't find the threshold concentration mentioned anywhere, which would be important to know. Does that concentration correlate with inhibition of formin activity by XopR? Maybe I missed it? Even so, the threshold concentration would be different for mobile and immobilized SLB.
4. In terms of wording, the authors sometimes use coacervate (coacervation) and sometimes condensate (condensation), are the authors use them interchangeably or mean to differentiate something? I feel it is the later, regarding the charge-driven phase separation. But I am not sure. The authors point out the negative charges are likely the driver for XopR phase separation. I want to point out there are positive charge patches in the protein as well and some work shows it's not the net charge but the charge pattern that is more important for phase separation (Das, J. Phys. Chem. B 2018, 122, 21, 5418–5431). Also even phase separation is affected by salt, it does not mean other interactions between IDR do not play a role in promoting phase separation, pi-pi interaction in some IDRs are also important for example (Vernon, eLife 2018; 7:e31486).
5. Page 4, two typos in 'Arabidopsis exhibited differentces'.

Reviewer #2 (Remarks to the Author):

In their paper entitled "Xanthomonas Effector XopR Hijacks Host Actin Cytoskeleton Via Complex Coacervation" Sun et al. analyze the mechanism by which effector proteins delivered via the TypeIII secretion system hijack host protein machineries to evade resistance. The focus here is on the actin cytoskeleton. There are many papers describing the host responses of the actin system to pathogens or immune receptor activation, but molecular insights into how the pathogen modifies that actin network are absent. This paper provides interesting data indicating that the effector XOPR self-aggregates in vivo, and these aggregates may recruit nucleators of the Formin class and they may bundle actin filaments directly. The paper combines live cell imaging of the time course of XOPR aggregation and the associated effects on cortical actin. The live cell imaging data are supported by a battery of biochemical assays of protein aggregation that are consistent with the idea that XOPR forms multivalent complexes to enable actin network modification. Overall the paper is novel, well designed, and carefully executed providing important new data that will have a broad impact on the actin cell biology field. Below are a some comments that would help to strengthen the authors conclusions.

- 1) My biggest concern is that the in vitro biochemical data are not strongly linked to the live cell

imaging that document rearrangement of the actin network. Are the actin phenotypes that are observed sensitive to Formin function? There are existing mutants and chemical inhibitors of formins. Are there XOPR mutants that do not have the expected effect? The authors may have better signal to noise in their estradiol system to compare mutant and wild type constructs if they were to follow the same cells over time, which is not an insurmountable technical challenge. XOPR is shown to promote bundling in vitro, but in vivo it does not localize to bundles. How is this discrepancy explained?

2) Figure 2 is highly technical and while clearly written it will be difficult for the general reader to follow the analyses and the significance of the data. It would be helpful to the reader to have some type of benchmark example quantifying the behaviors of a known high valency protein complex component or at least cite some comparative values to give these results meaningful context.,

Related: XOPR is a trimer at 50 mM NaCl. This does not explain the higher order aggregations that are depicted in the image data or SPR. Are there major peaks in the void that are not being discussed?

3) Along similar lines, the authors jump to a strong hit AtFH1-FH1C, as a potential interactor that is claimed to be passively recruited into the aggregate. There are no negative controls to demonstrate the specificity of this recruitment. AtFH1 mutant proteins should also be analyzed.

4) Can the authors explain the bleaching experiment in Fig. 3E? Why is there such a subtle effect on bleaching when so many FH6 molecules are claimed to be clustered by XOPR? What do the single cluster time-dependent bleaching traces look like?

5) This is a highly technical paper that is riddled with jargon and acronyms that make it annoying to read. Eliminate any acronym that is not absolutely essential.

Reviewer #3 (Remarks to the Author):

In this manuscript, Sun and coworkers investigate the role of the phyto-bacterial type III effector (T3E) XopR in promoting host subversion by *Xanthomonas campestris*. They identify a large intrinsically disordered region in XopR that undergoes liquid-liquid phase separation and promotes clustering of formins and enhanced actin network assembly at the cell cortex. The authors extensively characterize the mechanism of XopR self-assembly and its effects on actin filament nucleation, bundling and depolymerization. Their results support a model for a clever mechanism by which bacterial T3Es enable hijacking of the host cell's actin cytoskeleton and subversion of host biology.

The experimental results presented in this manuscript are well motivated and constitute a compelling advance in our understanding of the role of intrinsically disordered proteins in modulating actin cytoskeletal dynamics. I make a number of suggestions below largely aimed at increasing the level of quantitation of the data. I support publication of this paper once these points have been addressed.

(1) Figure 1b,d,e: Please plot these data on graphs with linear x-axis scales. This will enable readers

to assess the magnitude of the “burst” in actin filament bundling activity that occurs between 12 and 24 hpi.

(2) Figure 2b,g: Can a binding constant for XopR self-association and the association of AtFH1 with surface-immobilized XopR be obtained from these data? If not, can the curves be quantified and plotted as a function of XopR or AtFH1-FH1C concentration to allow the reader to interpret them?

(3) Figure 2c: The single elution peak for XopR observed at 50 mM NaCl (red curve) suggests a relatively homogeneous size for XopR multimers at this concentration. Is this interpretation correct? Also, how does the number of subunits in this multimeric complex compare to the size of the phase-separated droplets observed by microscopy in Figure 2d?

(4) Can the authors speculate on the nature of the interaction between XopR and AtFH1 FH1-C? Is it thought to be a specific interaction involving particular binding motifs in each protein (if so, which ones?), or is it a non-specific electrostatic interaction?

(5) I recommend that the authors include a description of type I formins to help orient readers who are unfamiliar with plant formins. For example, does AtFH1 contain a membrane-binding domain? What is its typical cellular localization pattern?

(6) Figure 3i: In the pyrene-actin assembly assay, the authors plot their data on a graph with a non-linear x-axis. I recommend plotting rates as a function of XopR concentration and indicating the concentration of AtFH1-FH1C to allow readers to calculate the protein ratios for themselves. Also, what is the predicted oligomeric state of XopR at these concentrations? How do these XopR concentrations compare to the binding constant for the XopR-AtFH1 interaction?

(7) Figure 5f,g: The Figure legend indicates that the data were fit using a Hill equation. It is my understanding that the fraction of actin in the pellet in low-speed co-sedimentation assays is not necessarily indicative of the number of binding sites that are occupied by XopR. For example, it is not known how many XopR are required to bind and bundle two actin filaments. A better way to fit these data would be to plot the concentration of bound XopR divided by the concentration of bundled actin on the y-axis instead. These data can then be fit with a model to obtain binding constants.

(8) Many formins are autoinhibited and require the binding of an activating protein (such as a Rho GTPase) to relieve the autoinhibition and begin assembling actin filaments. Can the authors speculate about how XopR-mediated co-localization might impact formin activation?

(9) WH2-like motifs are a common feature of the C-terminal domains of formins. For example, the formin INF2 contains a WH2-like motif that promotes both actin nucleation and filament turnover (Gurel 2015 JBC). Can the authors comment on the similarities and differences between the mechanisms of action of formins that encode their own WH2 motifs in cis, and the effects of association of a formin with a protein like XopR that contains a WH2-like motif in trans?

(10) Do WH2 motifs compete with profilin for binding to actin monomers? Given that profilin inhibits spontaneous actin nucleation, but enhances the filament elongation activity of formins, can the authors comment on the effects of WH2-mediated displacement of profilin from actin monomers on the nucleation and elongation activities of formin?

REVIEWER COMMENTS

Reviewer #1 (Remarks to the Author):

Interactions between intrinsically disordered regions (IDRs) are shown to drive phase separation for many proteins. Sun et. al. show that IDR-containing phyto bacterial type III effector (T3E) XopR hijacks the Arabidopsis actin cytoskeleton actin network in vivo by driving formin clustering and inhibiting depolymerization. The authors report that XopR IDR drives XopR phase separation and mediates actin polymerization and depolymerization in vitro by interacting with Formin, G-actin and F-actin. This work provides mechanistic insights on how T3E IDR help avoid host defense via phase separation. The most exciting discovery, to me, is that 'by tuning the physical-chemical properties of XopR-complex coacervates, XopR progressively manipulates multiple steps of actin assembly, including formin-mediated nucleation, crosslinking of F-actin, and actin depolymerization by competing for actin-depolymerizing factor, depending on constituent stoichiometry.' My slight disagreement would be what physical-chemical properties are turned and how. The authors provided a biphasic model for the regulation of formin activity by XopR where at early stage low concentration of XopR clusters formin to promote actin nucleation and at later stage high concentration of XopR inhibits formin by 'transforming the formin nanoclusters into condensed foci with higher-level clustering'. The main evidence is that at high XopR concentration, actin seed number with or without formin is the same (Fig 4B). The authors reason that 'the biphasic regulation of formin activity might indicate a change in the material properties from the gelation without phase transition to gelation plus phase separation'. It is not clear whether the critical concentration of phase separation is the same at which inhibition for formin activity starts.

We thank Reviewer 1 for these profound insights and constructive suggestions. We have now performed nano-scale clustering characterization by TIRF in Extended Data Figure 5g, in which 160 nM XopR induced the formations of many large clusters of FH1-FH1C (10 nM), a condition with abolished AtFH1 activities (also see below high G-actin concentration assay). Thus, the activities of 10 nM formin could be modulated by XopR at a nano-scale level starting from the 40-50 nM XopR (Extended Data Figure 5g). To be noted, the critical concentration of phase separation in Fig. 2e was characterized in bulk solution by wide-field microscopy with a less detection sensitivity, which usually starts to observe the clear droplets at the microscale. Therefore, the concentration of XopR in Fig. 2e is not practical to be directly adopted to interpret the formin activities in the TIRF assay that was carried out with the nanomolar proteins.

Given the curves in Fig 4B start to have a smaller slope at about 10mins at 160 nM XopR, I suspect other parameters are limiting the nucleation, maybe the amount of available G-actin. If enough free actin is available for both Formin and XopR, XopR+formin may still be more efficient at nucleating actin than XopR alone at high XopR concentration. The author can use higher G-actin at high XopR concentration to rule out this possibility.

We have now performed the suggested TIRF actin polymerization assay, which contains a higher concentration of G-act at 1.5 uM while keeping the AtFH-FH1C (10 nM) XopR (160 nM) at the same condition of the previously 0.5 uM G-actin assay. We did not find additional formin activities on top of the XopR-mediated nucleation, suggesting the abolished formin activities under such conditions, even with an excess of G-actin to be used. The new results are added now in the Extended Figure 6g-i. To be noted, such in vitro biochemical results are also consistent with in vivo results (Fig. 1f), where 24h Xcc infection inhibits actin nucleation when accumulated in the host at the late stage of infection by type-III secretion.

Even formin is inhibited, the seed number is still dramatically increased. This does not support XopR phase separation inhibits actin polymerization. XopR indeed directly nucleates actin, efficiently. An alternative interpretation is that high concentration of XopR is so effective at nucleating actin, the role of formin is not important anymore. Inhibition of polymerization at high XopR concentration could be attributed to binding of XopR to the polymerized actin.

We appreciate the excellent comments that improve the clarity of our manuscript. We agree with Reviewer's interpretations, and XopR indeed enhances actin nucleation directly shown in Figure 4d,e. Here, we have now clearly stated the XopR activity in nucleating actin on page 11. To test the scenario that Reviewer mentioned, we have performed the suggested nucleation assay using a higher concentration of G-actin (also mentioned in the above paragraph) and added new results in Extended Figure 6g-i. We have now added the new results and discussed in the revised manuscript.

The authors focus on the effect of XopR concentration change overtime, another parameter in the system that is changing overtime is the actin: from monomer, to polymer, to bindles. I see a continuum of phase separation from XopR with Formin to initiate actin nucleation, to XopR with G actin to further promote actin polymerization, to XopR with F actin to form bindles, which then inhibits actin depolymerization by forming a liquid to coat the actin and prevent access of depolymerization factors. In this multi-component phase separation system, IDR in XopR provides the weak transient interaction needed for phase separation, formin dimerization and location on membrane (reduced mobility) decrease concentration needed for XopR phase separation, which is further reduced by the nucleated actin polymer. Supporting this model, polymerization promotes phase separation by increasing valence and actin binding proteins form liquid coat on actin bindles have been reported (Sanders, Cell 2020 P306-324.E28., Weirich, PNAS 2017 114 (9) 2131-2136). ED Fig. 6K clearly show XopR coating on actin bindles and I even see larger foci in XopR channel that are not present in the actin channel. Can the authors follow the kinetics of actin nucleation to polymerization and to bindle in the system while observing XopR phase separation? That would help differentiate the two models.

We appreciate the excellent comments and the great insights.

Membrane-association decreases the concentration needed for XopR phase separation is important to understand the membrane-bounded membraneless organelle. Indeed, XopR proteins form small droplets on SLB at 150 mM (Extended Data Fig. 5k), though not visible in solution Fig. 2e at the same ionic strength. Now, we also added a description of this in the discussion on page 15.

To compare with the tactoid F-actin bundles by filamin (Weirich, PNAS 2017), we have now followed actin polymerization under TIRF over time. In the presence of Formin and XopR in vitro, actin filaments form liquid bundles over time into a tactoid shape, similar to the Gardel Lab reported using actin crosslinker filamin (Weirich, PNAS 2017). Our new results agree with the interpretation by the Reviewer. We have now added this result as Figure 5h and Supplementary video 3. In addition, we added more discussion on dose-dependent multiphase condensation and cited the Sanders, Cell 2020. We also discussed more on two-step inhibition of actin depolymerization: a) competing for ADF binding pocket at low concentration of XopR further; b) limiting ADF access by liquid coat F-actin bundles at a higher concentration, with a citation of Weirich, PNAS 2017 on Page 11.

If not, make sure there is excess G-actin for XopR in nucleation and compare the concentrations at which XopR phase separation happens and formin inhibition starts would be informative (on the same SLB).

Besides the above experiments mentioned in the above paragraph, we have also performed this suggested alternative assay using high concentration G-actin (Extended Figure 6g-i).

Other minor concerns are:

1. I don't see deletion of XopR IDR on effect of formin clustering or actin polymerization/depolymerization in cells, which will provide direct evidence for the functional role of IDR phase separation, if doable.

We have now compared the XopR-FL and XopR-C effect on formin clustering by performing in vivo tobacco transient expression by co-expressing XopR variants with AtFH6-GFP. It shows that while XopR-FL-expression immobilized formin and induced formin clustering on the plasma membrane, XopR-C without the N-terminal IDR could not target the cell surface did not introduce a noticeable change of formin dynamics. We added the new data as Extended Data Fig 4j-m.

2. I don't understand why the authors inhibit SLB fluidity. That would enhance protein phase separation. Is the PM less dynamic than SLB?

We apologize for the confusion. The immobilized SLB for single-particle FRAP assay was designed to study formin's bleaching trace over time in a single particle manner, in which dynamic-SLB is not practical to follow the formin signals during the period of bleaching and imaging acquisition. And such immobilized-SLB was only used for Fig. 3g, and Extended Data Fig5g,h to quantify the bleaching steps, whereas all the other SLBs are dynamic. Now we have indicated SLB conditions clearly in the figure legend to improve clarity.

3. 'Here, we used single-particle TIRF imaging to resolve the threshold concentration of complex coacervation of XopR-formin around the nucleation of phase transition by monitoring the dynamic fusion and clustering of individual formin puncta on the cell surface (Fig 3g and Extended Data Fig 5f)'. My understanding is 3g are from Extended Data 5G, which are immobilized SLB. While 5f is in mobile SLB and there is no XopR concentration noted. They are not on the cell surface.

We apologize for the wrong description. It is an SLB experiment. We have corrected that now. Extended Data Fig. 5f demonstrates the AtFH1 dynamics and fusion events on the SLB right after adding 100 nM XopR. We have now added the missing information and clearly described the experimental condition in the Figure legend.

I don't find the threshold concentration mentioned anywhere, which would be important to know. Does that concentration correlate with inhibition of formin activity by XopR? Maybe I missed it? Even so, the threshold concentration would be different for mobile and immobilized SLB.

We have now described the threshold concentration on page 9 based on the signal intensity change in ED 5j. Starting from 50 nM XopR at a XopR-formin stoichiometry 1:5, formins showed a clear up-shift in clustering, suggesting an approximate threshold concentration (Extended Data Fig 5j). In addition, the increasing concentration of XopR will eventually generate large micron-sized clusters (Extended Data Fig 5g). The effective concentration of XopR that clusters formin in vitro is consistent with its biphasic modulation of formin activities in actin nucleation in TIRF assay.

4. In terms of wording, the authors sometimes use coacervate (coacervation) and sometimes condensate (condensation), are the authors use them interchangeably or mean to differentiate something? I feel it is the later, regarding the charge-driven phase separation. But I am not sure.

Thanks for pointing out the consistency issue of nomenclature. We have used the word "coacervation" for the XopR mediated *in vitro* LLPS, in which electrostatic interaction is the underlying driving force. There are two places on page 9 where we used "condensates", which were meant to describe the morphology of *in vivo* XopR-formin clusters that we do not know their complex chemical-physical properties. These *in vivo* condensates are multi-component protein clusters in cells that might involve additional interaction forces that we do not understand yet, besides the primary electrostatic interaction. Using such different names in different contexts might help avoid misleading that *in vivo* "condensates" are only electrostatic. We want to be more careful before obtaining a holistic understanding of the *in vivo* interactome of XopR condensates.

The authors point out the negative charges are likely the driver for XopR phase separation. I want to point out there are positive charge patches in the protein as well and some work shows it's not the net charge but the charge pattern that is more important for phase separation (Das, J. Phys. Chem. B 2018, 122, 21, 5418–5431). Also even phase separation is affected by salt, it does not mean other interactions between IDR do not play a role in promoting phase separation, pi-pi interaction in some IDRs are also important for example (Vernon, eLife 2018; 7:e31486).

We have now revised the wording on Page 6 to avoid excluding potential contributing factors attributed to the XopR coacervation.

5. Page 4, two typos in 'Arabidopsis exhibited differences'.
We have corrected them now.

Reviewer #2 (Remarks to the Author):

In their paper entitled "Xanthomonas Effector XopR Hijacks Host Actin Cytoskeleton Via Complex Coacervation" Sun et al. analyze the mechanism by which effector proteins delivered via the Type III secretion system hijack host protein machineries to evade resistance. The focus here is on the actin cytoskeleton. There are many papers describing the host responses of the actin system to pathogens or immune receptor activation, but molecular insights into how the pathogen modifies that actin network are absent. This paper provides interesting data indicating that the effector XOPR self-aggregates *in vivo*, and these aggregates may recruit nucleators of the Formin class and they may bundle actin filaments directly. The paper combines live cell imaging of the time course of XOPR aggregation and the associated effects on cortical actin. The live cell imaging data are supported by a battery of biochemical assays of protein aggregation that are consistent with the idea that XOPR forms multivalent complexes to enable actin network modification. Overall the paper is novel, well designed, and carefully executed providing important new data that will have a broad impact on the actin cell biology field. Below are a some comments that would help to strengthen the authors conclusions.

1) My biggest concern is that the *in vitro* biochemical data are not strongly linked to the live cell imaging that document rearrangement of the actin network. Are the actin phenotypes that are observed sensitive to Formin function? There are existing mutants and chemical inhibitors of formins.

Thanks for the constructive suggestion about how to better support the formin functions underlying XopR mediated actin remodeling. XopR effects on multiple Type I Formin Arabidopsis, which has 11 homologs and likely has high functional redundancy. Thus, it is not practical at this moment for us to delete all formin homologs to generate a clean genetic background to address the point. Therefore, we took Reviewer's suggestion of using the "inhibitor". We applied formin inhibitor SMIFH2 to observe XopR-mediated actin remodeling. When SMIFH2 inhibitor was applied for 3h during first 6h period of XopR overexpression, we found that the XopR expression-enhanced actin polymerization was inhibited. We added the new data now in Extended Data Fig 1h,I, and described the results on page 5.

Are there XOPR mutants that do not have the expected effect? The authors may have better signal to noise in their estradiol system to compare mutant and wild type constructs if they were to follow the same cells over time, which is not an insurmountable technical challenge.

To answer this question, we have now performed tobacco transient expression that expresses different combinations of AtFH6-GFP with XopR-FL and XopR-C, respectively. As a result, while XopR-FL expression immobilized the formin dynamics and induced the formin clustering on the plasma membrane, XopR-C, without the N-terminal IDR, could not target the cell surface and did not introduce a noticeable change of formin dynamics. We added the new data as Extended Data Fig 4j-m and described the result on page 8.

XOPR is shown to promote bundling *in vitro*, but *in vivo* it does not localize to bundles. How is this discrepancy explained?

XopR has an N-terminal membrane-binding amphipathic helix (MBM, Fig 2a), which usually facilitates the membrane insertion and maintains a membrane-binding equilibrium between the membrane cytoplasm¹. Interaction with type-I formin also provides additional PM association of XopR. We think the bundling of F-actin *in vivo* could be derived from two resources: 1) The bundling of F-actin could be contributed by the clustering effect of the formins associated with F-actin barbed ends, which generate multiple F-actin filaments next to each other and easy to be coupled by

crosslinking factors. Our recent bioengineered multivalent-formin demonstrated the ability in developing localized node for actin crosslinking (Figure R1) ². 2) the cytosolic pool of XopR could also contribute to the F-actin bundling at the cell cortex, although low in concentration, such as at a submicromolar concentration (Fig. 5F), but sufficient for the sparse decoration along the F-actin filaments. Due to the low signal intensity for the sparsely-decorated XopR on the filament, it is technically challenging to visualize the fluorescent signal of the F-actin-decorated XopR by having strong nearby XopR signals on the PM. Now, we have now revised Fig.7 to reflect the above explanations and also discussed them in the discussion on page 16.

Figure R1. Representative images of TIRFM actin polymerization assay using clustered AtFH1C proteins (10% Alexa -647-labeled B-FH1C) on glass slides. 30 nM Biotin-tagged AtFH1C (magenta) was mixed with 3.75 nM streptavidin (S) before mixed with G-actin (containing 10% Oregon green 488-actin and 0.5% biotin-actin).

2) Figure 2 is highly technical and while clearly written it will be difficult for the general reader to follow the analyses and the significance of the data. It would be helpful to the reader to have some type of benchmark example quantifying the behaviors of a known high valency protein complex component or at least cite some comparative values to give these results meaningful context., Thanks for the suggestion. To facilitate understanding the biophysical parts with less technical jargon, we have now added some descriptions of different complex coacervation systems that could be compared with each other for an easier understanding. We added the following content on page 6. Due to high water content, many LLPS systems have very low interfacial energy³ compared to the classical oil-water interface, which has interfacial energy of ~ 60 mJ/m² (Israelachivli J.N., Intermolecular and Surface Forces, Academic Press, 3rd Edition). Charge-driven complex coacervate formed by polylysine (PLys) and polyglutamic acid (PGA) showed interfacial energy of smaller than 1 mJ/m², which is similar to the values measured in our study. Additionally, the interfacial energy of PLys-PGA coacervate also decreases as a function of solution salt concentration⁴. In addition, a comparable low interfacial energy (<1 mJ/m²) was also reported in mussel-inspired complex coacervate that is formed by mixing a recombinant mussel adhesive protein (fp-151-RGD) with hyaluronic acid³.

Related: XOPR is a trimer at 50 mM NaCl. This does not explain the higher order aggregations that are depicted in the image data or SPR. Are there major peaks in the void that are not being discussed?

XopR actually forms oligomers at 50 mM NaCl. The trimer formation is under 150 mM (Fig. 2c). Nevertheless, neither 150 mM nor 50 mM NaCl condition results in a void peak. Void peak usually indicates protein aggregates, while XopR proteins are soluble high-order oligomers that enable LLPS;

The void peak protein sample would usually show amorphous assemblies signals under the microscope, which would look different from the droplets shown in Fig. 2d.

Our SPR and fluorescent imaging were both done at 150 mM. Under such condition, XopR proteins form homogenous trimers (Fig. 2). At 150 mM, a trimerization of XopR (Fig. 2c) is necessary but not sufficient in driving a robust coacervation. The high-order assembly of XopR (Fig. 2d) requires additional electrostatic interactions between the positively and negatively charged patches (Fig. 2a). A similar scenario of multi-binding effects for phase separation is the stress granule study from Clifford Brangwynne lab, in which the dimerization RNA binding protein G3BP is not sufficient to drive the formation of stress granule, unless forming a high valency complex of G3BP-UBAP2L that provides more RNA binding-interfaces⁵.

3) Along similar lines, the authors jump to a strong hit AtFH1-FH1C, as a potential interactor that is claimed to be passively recruited into the aggregate. There are no negative controls to demonstrate the specificity of this recruitment. AtFH1 mutant proteins should also be analyzed.

We apologize for the confusion in formin specificity. Actually, we also do not think AtFH1 is a specific formin member to be targeted by XopR. Our results suggest a general mechanism for type-I formin isoforms, because the functional formin-homology 2 (FH2) domains are highly conserved among these formins. By having the conserved domains of all type I formins at the cell surface, where XopR also localizes, we think XopR could interact with multiple formin homologs. We have demonstrated that both AtFH1 and AtFH6 interact with XopR (Fig. 2g and Extended Data Fig 4c,d). AtFH1 is a well-characterized formin representative for biochemical experiments⁶, and AtFH6-GFP is a ready-available stable transgenic plant to use as another representative for in vivo study^{7,8}. We have now clarified this in the results to avoid confusion.

4) Can the authors explain the bleaching experiment in Fig. 3E? Why is there such a subtle effect on bleaching when so many FH6 molecules are claimed to be clustered by XOPR? What do the single cluster time-dependent bleaching traces look like?

To describe better our photobleaching methods, we have now added representative images of single-particle images before and after bleaching and the bleaching traces in Extended Data Fig. 4a. To clarify the effective clustering of Formin by XopR, we have now indicated the accumulative ratio change in formin bleaching steps after Xcc infection by normalizing the sum of the bleaching steps of mock. The new analysis was now added in Fig. 3e and in the legend to indicate overall changes of formin-clustering with XopR in photobleaching assay.

5) This is a highly technical paper that is riddled with jargon and acronyms that make it annoying to read. Eliminate any acronym that is not absolutely essential.

We apologize for the readability issue. We have now carefully checked throughout the manuscript and tried to avoid acronyms if it is possible. For terms are often repeated, we provided the acronyms' full name at the beginning of each session of introduction, results, and discussion.

Reviewer #3 (Remarks to the Author):

In this manuscript, Sun and coworkers investigate the role of the phyto-bacterial type III effector (T3E) XopR in promoting host subversion by *Xanthomonas campestris*. They identify a large intrinsically disordered region in XopR that undergoes liquid-liquid phase separation and promotes clustering of formins and enhanced actin network assembly at the cell cortex. The authors extensively characterize the mechanism of XopR self-assembly and its effects on actin filament nucleation, bundling and depolymerization. Their results support a model for a clever mechanism by which bacterial T3Es enable hijacking of the host cell's actin cytoskeleton and subversion of host biology.

The experimental results presented in this manuscript are well motivated and constitute a compelling advance in our understanding of the role of intrinsically disordered proteins in modulating actin cytoskeletal dynamics. I make a number of suggestions below largely aimed at increasing the level of quantitation of the data. I support publication of this paper once these points have been addressed.

(1) Figure 1b,d,e: Please plot these data on graphs with linear x-axis scales. This will enable readers to assess the magnitude of the "burst" in actin filament bundling activity that occurs between 12 and 24 hpi.

We have revised the x-axis with a linear scale now, as suggested.

(2) Figure 2b,g: Can a binding constant for XopR self-association and the association of AtFH1 with surface-immobilized XopR be obtained from these data? If not, can the curves be quantified and plotted as a function of XopR or AtFH1-FH1C concentration to allow the reader to interpret them? SPR results have K_{on1} (on-rate constant/association rate constant, $M^{-1} s^{-1}$) and K_{off1} (off-rate constant/dissociation rate constant, s^{-1}), from which we can base on the pseudo-first-order kinetics to get a dissociation constant (K_{d1}), during the association phase. Currently, we do not know the explicit number of multivalent interaction sites for the inter- and intramolecular interactions within the complex coacervates of formin-XopR. Therefore, we are not able to claim the exact binding constant any further beyond the pseudo-first-order kinetics. Here, as suggested by the Reviewer, we have now added these values to the SPR sensorgram to the revised Fig. 2b,g to allow readers to interpret.

(3) Figure 2c: The single elution peak for XopR observed at 50 mM NaCl (red curve) suggests a relatively homogeneous size for XopR multimers at this concentration. Is this interpretation correct? XopR shows multimers at 50 mM. Multivalent interaction proteins are usually heterogeneous in oligomerization states. The elution peak is close to, but not an avoid peak. This is because of the chromatographic column resolution's intrinsic limitation for separating the void peak from the macromolecular protein complex clearly. Due to the large size, the macromolecular heterogeneous protein complex is usually eluted by a single peak on FPLC from most commonly used columns, such as Superdex 200 and Sepharose 6 that have distinct ranges of resolutions to resolve proteins at different sizes.

Also, how does the number of subunits in this multimeric complex compare to the size of the phase-separated droplets observed by microscopy in Figure 2d?

Thanks for this excellent question. In theory, once the oligomer-core is formed, additional weak multivalent interactions will increase the valency and expand the network to generate phase separation. At 150 mM, a trimerization of XopR is necessary but not sufficient in driving a robust coacervation. The high-order assembly of XopR requires additional electrostatic interactions

between the positively and negatively charged patches, such as the multimers at 50 mM (Fig. 2c,d). It is reminiscent a stress granule study from Clifford Brangwynne lab, in which the dimerization RNA binding protein G3BP is not sufficient to drive the formation of stress granule unless forming a high valency complex of G3BP-UBAP2L that provides more RNA binding-interfaces⁵. Once the critical concentration of the LLPS is reached (Fig. 2e), the initial demixing droplets start to form, basing on the binodal model in thermodynamics. Afterward, the droplet could fuse and grow over time without changing the valency towards thermodynamic phase coexistence. Thereby, the droplet size in Fig. 2d does not reflect the LLPS properties but rather a snapshot during the phase growth at the liquid status. If the solution-to-gel or gel-to-solid transition happens over time, it is because of the increase of bulk valency in a material properties-dependent manner. Therefore, there is no single answer for a quantitative correlation between droplet size and protein valency because of the interdependence and co-evolving feature between the droplets' physicochemical properties and size. At this moment, we do not have appropriate techniques to quantitatively determine XopR valency following their phase growth with size increase.

(4) Can the authors speculate on the nature of the interaction between XopR and AtFH1 FH1-C? Is it thought to be a specific interaction involving particular binding motifs in each protein (if so, which ones?), or is it a non-specific electrostatic interaction?

Thanks for raising this excellent question. XopR self-conservation is electrostatic-dependent that is validated by several assays. For example, XopR-scGFP (-30) complex coacervates showed greater adhesive forces than XopR-coacervates in the SFA experiment, further supporting electrostatic dependent interaction. However, the SFA experiment did not show a noticeable difference in surface tension between XopR and AtFH1 FH1-C complex coacervates and XopR self-conservates. Therefore, the interaction forces between XopR and AtFH1-FH1C are mainly driven by other interactions independent of electrostatics. Currently, we do not have explicit information on the interaction types and interaction position within the XopR-AtFH1-FH1C complex. Multiple types of interactions could contribute IDR mediated phase separation, such as hydrogen-bond and pi-pi interaction in IDRs^{9,10}. Our ongoing collaborative-efforts are trying to resolve the interaction information by NMR approaches, such as our collaborator published recently about intrinsically disordered histidine-rich squid beak proteins (HBPs)¹¹. Here, we have now added the information about the non-electrostatic nature between XopR and Formin to the manuscript on page 7.

(5) I recommend that the authors include a description of type I formins to help orient readers who are unfamiliar with plant formins. For example, does AtFH1 contain a membrane-binding domain? What is its typical cellular localization pattern?

Type I formin all localize on the plasma membrane by having a transmembrane domain. We have now added more information of type I formin on page 7 as suggested.

(6) Figure 3i: In the pyrene-actin assembly assay, the authors plot their data on a graph with a non-linear x-axis.

I recommend plotting rates as a function of XopR concentration and indicating the concentration of AtFH1-FH1C to allow readers to calculate the protein ratios for themselves.

In this part, we conclude that XopR-formin stoichiometry is critical to tune the formin activity. Therefore we feel the stoichiometry-based indication on X-axis would present our data better. As the Reviewer suggested for easy reference, we have now provided the protein concentrations in the figure legend on page 31.

Also, what is the predicted oligomeric state of XopR at these concentrations?

Fig 3i experiments were performed with 150 mM NaCl, in which XopR is the trimers on its own. When XopR interacts with Formin, XopR starts to cause higher valency for the XopR-formin complex and generate higher-oligomerization of formin starting from 40 nM (Extended Data Fig 5h-j), and even higher oligomerization of the complex by additional XopR such as until the concentration of 160 nM at a formin:XopR stoichiometry of 1:16 (Extended Data Fig 5h-j). The oligomerization states of XopR are evolving in a stoichiometry-dependent manner. Now we have added more discussion on such stoichiometry-dependent change in oligomerization states on page 14.

How do these XopR concentrations compare to the binding constant for the XopR-AtFH1 interaction?

As mentioned in the previous explanations, we do not know the binding site information, such as how many interaction sites within the formin-XopR complex, and thereby we cannot claim the exact binding constant any further beyond the pseudo-first-order kinetics. SPR has K_{on1} (on-rate constant/association rate constant, $M^{-1} s^{-1}$) and K_{off1} (off-rate constant/disassociation rate constant, s^{-1}), we can base on the pseudo-first-order kinetics to get dissociation constant (K_{d1}), during the association phase. Here, we have now added these values to the SPR sensorgram to the revised Fig. 2b,g to allow readers to interpret. By undergoing the LLPS, the valency within the XopR-AtFH1 complex likely to be enhanced, which might change binding kinetics through changing the proximity-induced effective-concentration in a manner of noncanonical binding dynamics¹². We currently do not know the physical pattern and connectivity of the XopR and AtFH1 in the droplets. We hope our ongoing NMR structural studies will give a better answer soon.

(7) Figure 5f,g: The Figure legend indicates that the data were fit using a Hill equation. It is my understanding that the fraction of actin in the pellet in low-speed co-sedimentation assays is not necessarily indicative of the number of binding sites that are occupied by XopR. For example, it is not known how many XopR are required to bind and bundle two actin filaments. A better way to fit these data would be to plot the concentration of bound XopR divided by the concentration of bundled actin on the y-axis instead. These data can then be fit with a model to obtain binding constants.

Sorry for the confusion. In this experiment, we were not intended to calculate the binding sites of XopR on F-actin. Here we aim to understand the minimum XopR concentration that provides effective F-actin crosslinking and examine the potential ionic strength-dependency. Now, we have clarified this point in the corresponding result part on Page 11.

(8) Many formins are autoinhibited and require the binding of an activating protein (such as a Rho GTPase) to relieve the autoinhibition and begin assembling actin filaments. Can the authors speculate about how XopR-mediated co-localization might impact formin activation?

The type-I formin has N-terminal facing outside of the cell and C-terminal FH1 and FH2 in the cytoplasm, which does not have the autoinhibition mechanism of mammalian formin homologs, whose activation requires the binding by GTPase proteins⁷. In plants, the plasma membrane anchoring of type-I formin proteins constrained their motility and localization. How type-I plant formin activity is regulated was unknown. Here, our XopR works suggest a mechanism by which formin activity could be regulated through molecular condensation. To clarify this point, we have now added more discussion on page 13.

(9) WH2-like motifs are a common feature of the C-terminal domains of formins.

For example, the formin INF2 contains a WH2-like motif that promotes both actin nucleation and filament turnover (Gurel 2015 JBC).

INF2 is a unique member of the formin family with several biochemical activities and has a C-terminal WH2 motif. The C-terminus of Arabidopsis formin is the FH2 domain without additional C-terminal flexible sequences for yeast or mammalian formins¹³. By analyzing 21 Arabidopsis formins, we could not identify any potential WH2 motif at their C-terminus.

Can the authors comment on the similarities and differences between the mechanisms of action of formins that encode their own WH2 motifs in cis, and the effects of association of a formin with a protein like XopR that contains a WH2-like motif in trans?

- a) As mentioned above, we did not identify any potential WH2-like motif in Arabidopsis formins.
- b) Actin nucleation by formin and WH2-containing proteins is different by having distinct contact modes between nucleator and G-actin^{14,15}. We are not aware of any previous report showing synergistic promotion in actin nucleation by WH2 and FH2 domains.
- c) Based on our proposed working model for XopR and Formin, weak multivalent XopR at low-order oligomerization recruits G-actin to the formin FH2 domain, thereby promoting formin-mediated actin nucleation by increasing the local actin concentration. However, formin activity is inhibited at a high-order oligomerization complex state within XopR-formin complex coacervates due to the impaired conformational flexibility of formin in the nanoclusters. Now, we have added such explanation in the discussions on page 14.

(10) Do WH2 motifs compete with profilin for binding to actin monomers? Given that profilin inhibits spontaneous actin nucleation, but enhances the filament elongation activity of formins, can the authors comment on the effects of WH2-mediated displacement of profilin from actin monomers on the nucleation and elongation activities of formin?

Thanks for bringing up this interesting point. Theoretically, we believe that the WH2 motif could not directly compete for actin away from the profilin. Because the affinity between WH2 domain and G-actin (>16 μM in Kd) is lower by two orders of magnitude than the association between G-actin and profilin ($\sim 0.1 \mu\text{M}$ in Kd). Instead of competition, XopR likely facilitates formin nucleation by engaging more profilin-actin for the formin's FH1 domain. Now we have added this in the discussion on page 14.

- 1 Wieprecht, T., Apostolov, O., Beyermann, M. & Seelig, J. Thermodynamics of the alpha-helix-coil transition of amphipathic peptides in a membrane environment: implications for the peptide-membrane binding equilibrium. *J Mol Biol* **294**, 785-794, doi:10.1006/jmbi.1999.3268 (1999).
- 2 Ma, Z. *et al.* Formin Nano-clustering Mediated Actin Assembly during Plant flagellin and DSF-signalings. *Cell Rep* (**Accepted**) (2021).
- 3 Hwang, D. S. *et al.* Viscosity and interfacial properties in a mussel-inspired adhesive coacervate. *Soft Matter* **6**, 3232-3236, doi:10.1039/C002632H (2010).
- 4 Priftis, D., Laugel, N. & Tirrell, M. Thermodynamic characterization of polypeptide complex coacervation. *Langmuir* **28**, 15947-15957, doi:10.1021/la302729r (2012).
- 5 Sanders, D. W. *et al.* Competing Protein-RNA Interaction Networks Control Multiphase Intracellular Organization. *Cell* **181**, 306-324 e328, doi:10.1016/j.cell.2020.03.050 (2020).

- 6 Michelot, A. *et al.* The formin homology 1 domain modulates the actin nucleation and bundling activity of Arabidopsis FORMIN1. *Plant Cell* **17**, 2296-2313, doi:10.1105/tpc.105.030908 (2005).
- 7 Favery, B. *et al.* Arabidopsis formin AtFH6 is a plasma membrane-associated protein upregulated in giant cells induced by parasitic nematodes. *Plant Cell* **16**, 2529-2540, doi:10.1105/tpc.104.024372 (2004).
- 8 Van Damme, D., Bouget, F. Y., Van Poucke, K., Inze, D. & Geelen, D. Molecular dissection of plant cytokinesis and phragmoplast structure: a survey of GFP-tagged proteins. *Plant J* **40**, 386-398, doi:10.1111/j.1365-313X.2004.02222.x (2004).
- 9 Vernon, R. M. *et al.* Pi-Pi contacts are an overlooked protein feature relevant to phase separation. *Elife* **7**, doi:10.7554/eLife.31486 (2018).
- 10 Alberti, S., Gladfelter, A. & Mittag, T. Considerations and Challenges in Studying Liquid-Liquid Phase Separation and Biomolecular Condensates. *Cell* **176**, 419-434, doi:10.1016/j.cell.2018.12.035 (2019).
- 11 Gabryelczyk, B. *et al.* Hydrogen bond guidance and aromatic stacking drive liquid-liquid phase separation of intrinsically disordered histidine-rich peptides. *Nat Commun* **10**, 5465, doi:10.1038/s41467-019-13469-8 (2019).
- 12 Errington, W. J., Bruncsics, B. & Sarkar, C. A. Mechanisms of noncanonical binding dynamics in multivalent protein-protein interactions. *Proc Natl Acad Sci U S A* **116**, 25659-25667, doi:10.1073/pnas.1902909116 (2019).
- 13 van Gisbergen, P. A. & Bezanilla, M. Plant formins: membrane anchors for actin polymerization. *Trends Cell Biol* **23**, 227-233, doi:10.1016/j.tcb.2012.12.001 (2013).
- 14 Otomo, T. *et al.* Structural basis of actin filament nucleation and processive capping by a formin homology 2 domain. *Nature* **433**, 488-494, doi:10.1038/nature03251 (2005).
- 15 Benanti, E. L., Nguyen, C. M. & Welch, M. D. Virulent Burkholderia species mimic host actin polymerases to drive actin-based motility. *Cell* **161**, 348-360, doi:10.1016/j.cell.2015.02.044 (2015).

REVIEWERS' COMMENTS

Reviewer #1 (Remarks to the Author):

The authors have addressed all of my concerns.

Reviewer #2 (Remarks to the Author):

All of my concerns have been adequately addressed.

Reviewer #3 (Remarks to the Author):

The authors have addressed all of my comments, and I support publication of this manuscript.

Reviewer #4 (Remarks to the Author):

(LINE 178) The decrease in F_{ad} is likely due to the increase in the screening effect by the electrostatic interaction between XopR molecules.

-- Authors say that F_{ad} decreases with increasing salt concentration due to the electrostatic screening effects. It will be helpful, if authors can evaluate the decay length upon approach, and see if the decay length matches the theoretical Debye length.

(Line 186) coacervates exhibited greater adhesion forces ($E_{ad} = -12.9$ mN/m) and surface tension... ..

-- How did the authors evaluate E_{ad} ? Is it based on the Derjaguin approximation? Also, it will be good to explicitly write surface tension as well.

(Line 791) F_{ad} is correlated to the effective interfacial energy E by $E = -F_{ad}/4\pi R$. mN/m is equivalent to mJ/m².

-- Is this E different from E_{ad} that authors mention in Line 186? I suppose E_{ad} is from Derjaguin's approx., while E is identical to surface tension, γ , which has following relation

$$E_{ad} = -F_{ad}/2\pi R = 2E \rightarrow E = E_{ad}/2$$

E for effective interfacial energy is confusing. Designating γ or γ_{eff} , should be much suitable for the readers.

-- Starting a sentence with "mN/m", is not recommendable.

(Online methods) coacervate system can be a slowly changing dynamic system (Figure 6 of Priftis et al., Langmuir, 2012, 28, 8721-8729); So, it is important to mention approach and separation velocity.

Point to Point Response to Reviewers' Comments

We would like to thank the fourth reviewer for the constructive comments and recommendations. We have carefully considered these comments, and have made revisions in the manuscript accordingly. Changes we made in the revised manuscript are labelled in red.

Reviewer #4 (Remarks to the Author):

Comment 1. (LINE 178) The decrease in Fad is likely due to the increase in the screening effect by the electrostatic interaction between XopR molecules.

-- Authors say that Fad decreases with increasing salt concentration due to the electrostatic screening effects. It will be helpful, if authors can evaluate the decay length upon approach, and see if the decay length matches the theoretical Debye length.

Response:

For a monovalent electrolyte at 25 ° C (298K) the Debye length of aqueous solutions is (Israelachvili J N. Intermolecular and surface forces[M]. Academic press, 2015):

$$\begin{aligned}\frac{1}{\kappa} &= \left(\frac{\epsilon_0 \epsilon k T}{2 \rho_{\infty} e^2} \right)^{\frac{1}{2}} = \left(\frac{8.854 \times 10^{-12} \times 78.4 \times 1.381 \times 10^{-23} \times 298}{2 \times 6.022 \times 10^{26} \times (1.602 \times 10^{-19})^2 M} \right)^{\frac{1}{2}} = \frac{0.304 \times 10^{-9}}{\sqrt{M}} \text{ m} \\ &= \frac{0.304}{\sqrt{M}} \text{ nm}\end{aligned}$$

We have calculated the Debye length of all the salt conditions. In all our conditions, the decay lengths are close to or below 1 nm. The repulsion in our SFA force distance profile is mainly due to the steric repulsion of the proteins, not electrostatic interactions. The small decay length indicates that increasing the salt concentration can effectively screen the electrostatic interactions between the protein molecules, and lead to the decrease of the interfacial tension of the XopR coacervate. The same phenomenon has been reported in polypeptide based (poly(l-lysine hydrochloride) (PLys) and poly(l-glutamic acid sodium salt) (PGA)) complex coacervate system (Priftis et al. Langmuir 2012, 28, 23, 8721–8729).

C_{NaCl} (mM)	Debye length (nm)
50	1.36
100	0.96
125	0.86
150	0.78
200	0.68

Comment 2. (Line 186) coacervates exhibited greater adhesion forces ($E_{ad}=-12.9$ mN/m) and surface tension... ..

-- How did the authors evaluate E_{ad} ? Is it based on the Derjaguin approximation? Also, it will be good to explicitly write surface tension as well.

Response: We apologize that E_{ad} in Line 186 is a typo and it should be F_{ad} . The text and figure are all correct. XopR-scGFP(-30) coacervates exhibited greater adhesion forces ($F_{ad}= -12.9$ mN/m) which corresponding to a low interfacial energy of ~ 1.02 mN/m at 50 mM NaCl. We did not use Derjaguin approximation in our calculation. We evaluated the surface tension by the capillary force equation, $\gamma_{eff} = -F_{ad} / 4\pi R$ from ref 27. The typo has been corrected in the updated text.

Reference:

Ref 27 in Manuscript: Hwang, D.S. et al. Viscosity and interfacial properties in a mussel-inspired adhesive coacervate. *Soft Matter* 6, 3232-3236 (2010).

Comment 3. (Line 791) F_{ad} , is correlated to the effective interfacial energy E by $E = -F_{ad}/4\pi R$. mN/m is equivalent to mJ/m².

-- Is this E different from E_{ad} that authors mention in Line 186? I Suppose E_{ad} is from Derjaguin's approx., while E is identical to surface tension, γ , which has following relation

$$E_{ad} = -F_{ad}/2\pi R = 2E \rightarrow E = E_{ad}/2$$

E for effective interfacial energy is confusing. Designating γ or γ_{eff} , should be much suitable for the readers.

-- Starting a sentence with "mN/m", is not recommendable.

Response: As we have replied in the previous comment. We apologize that E_{ad} in Line 186 (comment #2) is a typo and it should be F_{ad} , and this typo has been corrected. The E in Line 791 is the effective interfacial Energy, $\gamma_{eff} = -F_{ad}/4\pi R$. We have clarified this in Line 792 on Page 25 as "The measured pulling force, F_{ad} , is correlated to the effective interfacial energy γ_{eff} by $\gamma_{eff} = -F_{ad} / 4\pi R$. The unit of γ_{eff} is mN/m which is equivalent to mJ/m²." We have also updated the y-axis label of Fig 2i to change 'Energy' to 'Effective interfacial energy'.

Comment 4. (Online methods) coacervate system can be a slowly changing dynamic system (Figure 6 of Priftis et al., *Langmuir*, 2012, 28, 8721-8729); So, it is important to mention approach and separation velocity.

Response: We appreciate and fully agree with the reviewer that the approach and separation speed can affect the adhesion force measured. In the revised methods session on page 25 we have included the approach and separation speed. "The approach and separation speed of our SFA measurements was about 5 nm/s."